# Measurement report: New Particle Formation Events Observed during the COALA-2020 Campaign

Jhonathan Ramirez-Gamboa[1,2], Clare Paton-Walsh[1*], Melita Keywood[2], Ruhi Humphries[2], Asher Mouat[4], Jennifer Kaiser[4,5], Malcom Possell[3], Jack Simmons[1], Travis A. Naylor[1]

[1] Centre for Atmospheric Chemistry, School of Earth, Atmospheric and Life Sciences, University of Wollongong, NSW 2522, Australia

[2] Climate Science Centre, CSIRO Environment, Aspendale, VIC 3195, Australia

[3] School of Life and Environmental Sciences, University of Sydney, NSW 2006, Australia

[4] School of Civil and Environmental Engineering, Georgia Institute of Technology, Atlanta GA 30332, USA

[5] School of Earth and Atmospheric Sciences, Georgia Institute of Technology, Atlanta GA 30332, USA

* Correspondence: clarem@uow.edu.au

**Abstract:**

Aerosols play an important role in atmospheric processes influencing cloud formation, scattering and absorbing solar radiation, and by affecting trace gases through chemical reactions occurring in and on aerosol particles. Ultimately aerosols affect the radiative balance of the earth modifying climate. A large fraction of aerosols is formed through chemical reactions following gas-to-particulate processes in the atmosphere: nucleation and growth. Biogenic Secondary Organic Aerosols (BSOA) are formed when plant produced volatile organic compounds (VOCs) react in the atmosphere through gas-phase oxidation. One of the highest BVOC emitting regions in the world is South-east Australia due to the high density of *Eucalyptus* species. The COALA-2020 (Characterizing Organics and Aerosol Loading over Australia) campaign worked towards a better understanding of biogenic VOCs in quasi-pristine conditions in the atmosphere and their role in particle formation.

The observations showed a highly reactive atmosphere with frequent new particle formation (NPF) occurring (42% days with data) often associated with pollution plumes. Analysis of NPF events suggested that $SO_2$ plumes likely triggered particle formation, while particle growth depended on available VOCs, hydroxyl radicals and the presence of multiple $SO_2$ intrusions promoted growth of smaller clusters. Nighttime NPF events coincided with monoterpene ozonolysis but were rare. These findings highlight the significant role of biogenic VOCs, in driving NPF and SOA formation in South-east Australia. The COALA-2020 campaign provided valuable insights into local atmospheric chemistry and its potential impact on regional air quality and climate. However, longer-term observations are crucial to understand seasonal variations, trends and extreme events.

**Keywords: COALA-2020; aerosols, BVOCs, NPF.**

## 1. Introduction

Aerosols can influence our health (Annesi-Maesano et al., 2013; Shi et al., 2016) but also play an important role in regulating Earth's energy balance, the hydrological cycle and even the abundance of key chemical species in the atmosphere such as hydroxyl radical (OH) and indirectly greenhouse gases (e.g., Kerminen et al., 2012). The chemical composition, size and concentrations of aerosols determine the effects on health

and the environment (Liu et al., 2016b; Pope and Dockery, 2006; Ren et al., 2017). Aerosols can be directly    40
emitted (primary aerosols) or they can be product of chemical reactions in the atmosphere (secondary    41
aerosols) (Pöschl, 2005).    42

Secondary aerosols are produced via gas-to-particle transition. New Particle Formation (NPF) occurs when    43
multiple reactions in the atmosphere create stable molecular clusters. Once the clusters are formed, they    44
can grow through coagulation and condensation potentially resulting in cloud condensation nuclei (CCN)    45
(Dal Maso et al., 2005; Hussein et al., 2005; Kulmala et al., 2001). Multiple factors determinate NPF in the    46
atmosphere including atmosphere composition and boundary conditions (temperature, humidity, planetary    47
boundary layer (PBL) height, turbulence) (Bousiotis et al., 2021; Wu et al., 2021; Xu et al., 2021a). Sulfuric    48
acid ($H_2SO_4$) is one of the main drivers of the nucleation process in the continental boundary layer, but it    49
does not explain all growth and nucleation rates (Sihto et al., 2006). The presence of ammonia ($NH_3$), amines    50
or ions in the atmosphere can enhance $H_2SO_4$ nucleation rates (Kirkby et al., 2023; Zhao et al., 2011; Zheng    51
et al., 2015). High levels of $SO_2$ and Volatile Organic Compounds (VOCs) will enhance NPF (Nestorowicz et    52
al., 2018; Song et al., 2019, p.20; Xu et al., 2021b).    53

VOCs are a group of carbon-based gases emitted by biological and anthropogenic sources that are    54
characterised by their high vapour pressure (Goldstein and Galbally, 2007; Kesselmeier and Staudt, 1999;    55
Matsui, 2006). VOCs can undergo hydroxyl radical (OH), ozone or nitrate radical ($NO_3$) oxidation in the gas    56
phase, producing compounds of varying volatilities, and products with low enough volatility can contribute    57
to NPF or partition to existing particles, resulting in particle growth.    58

The most common biogenic VOC (BVOC) is isoprene followed by monoterpenes. BVOCs play an important    59
role in secondary organic aerosol (SOA) formation (e.g., Mahilang et al., 2021). VOCs have been associated    60
with particle growth (Riipinen et al., 2012) but their role and the autoxidation mechanism was not    61
understood until recently (Bianchi et al., 2019). Autoxidation of monoterpenes supports the particle growth    62
process by generating highly oxygenated molecules (HOMs) via the formation of peroxy radicals (Bianchi et    63
al., 2019; Kirkby et al., 2023; Lehtipalo et al., 2018). HOMs can be characterised as ultra-low VOCs (ULVOC)    64
or extremely low VOCs (ELVOC) depending upon the saturation concentration (Bianchi et al., 2019; Peräkylä    65
et al., 2020).    66

Oxidation of monoterpenes is a significant pathway for SOA formation, yielding higher amounts of low-    67
volatility molecules like ULVOCs and ELVOCs compared to isoprene oxidation (Friedman and Farmer, 2018;    68
Lee et al., 2023; Luo et al., 2024; Riva et al., 2019; Zhang et al., 2018). HOMs are key precursors for new    69
particle formation. However, the atmospheric production of HOMs can be limited by competing reactions    70
and the presence of other VOCs. For instance, as a general principle, once a VOC molecule oxidizes, it    71
becomes more complex and forms larger Oxygenated VOCs (OVOCs) that are less likely to undergo further    72
oxidation, especially in the presence of other VOCs with higher reactivity towards OH or $O_3$ (Kiendler-Scharr    73
et al., 2009). An example of this limitation is the suppression of monoterpene-derived HOM formation by    74
isoprene oxidation products. These products can interfere with the formation of $C_{20}$ dimers from    75
monoterpene oxidation, leading to a reduced yield of HOMs and favoring the formation of weaker nucleating    76
species $C_{15}$ (Dada et al., 2023; Heinritzi et al., 2020; Liu et al., 2016a). This suppression effect is dynamic,    77
varying non-linearly with local atmospheric composition (e.g., isoprene and monoterpene concentrations,    78

oxidant availability) and atmospheric conditions (e.g., temperature, humidity, stability), which ultimately determine the dominant SOA formation pathways (e.g. Song et al., 2019).

Understanding BVOC emissions and their role in SOA formation is important to accurately predict aerosol properties and their impact on climate. However, BVOC are poorly characterized under Australian conditions (Paton-Walsh et al., 2022). MEGAN (The Model of Emissions of Gases and Aerosols from Nature) emissions show south-east Australia as one of the BVOC hot-spots in the region (Guenther et al., 2012) but multiple modelling studies have shown that MEGAN emissions estimation might not represent local conditions correctly in this region (Emmerson et al., 2016, 2018, 2019). Most of the Australian forested regions are dominated by high emitting *Eucalyptus* species (ABARES, 2019; Aydin et al., 2014; Padhy and Varshney, 2005) that combined with periods of high temperature and drought stress create the conditions for high emissions/concentrations of BVOCs in the atmosphere (Emmerson et al., 2020; Fini et al., 2017; Ormeño et al., 2007). The emissions ratios of isoprene to other VOCs are poorly constrained and the local chemistry is not well understood.

The COALA-2020 campaign worked towards a better understanding of biogenic VOCs in quasi-pristine conditions in the atmosphere and their role in local atmospheric chemistry in south-east Australia. COALA-2020 was a collaborative effort between local institutions including the University of Wollongong, CSIRO, ANSTO, and the University of Sydney, and international peers from Georgia Institute of Technology, The University of California, Irvine, Nagoya University and Lancaster University. This part of the study focused on identifying and characterising NPF events after the "Black Summer" 2019-2020 Australian bushfire season. Here we focus on identifying drivers and conditions in which NPF started or were enhanced in the local environment.

## 2. Materials and Methods

### 2.1 The COALA-2020 Campaign

The COALA-2020 campaign took place at Cataract Scout camp (34°14'44" S, 150°49'26" E) located 20 km north-northwest of Wollongong on the east coast of NSW, Australia. The site is surrounded by a heavily forested area mainly stocked by Eucalyptus species (see Figure 1). North of the sampling site is a four-lane arterial road connecting the M1 motorway on the east coast with south-western Sydney. Other possible anthropogenic sources impacting the site are two underground coal mine heads, located 1.5 km to the northeast and 2.5 km to the north). Further afield sources include the Sydney suburban area (around 18 km north-west), Sydney city (45 km north), Wollongong urban area and Port Kembla steelworks in the southern part of Wollongong (28 km to the southeast).

The campaign was conducted from the 17th January to the 23rd March, 2020. The first period of the campaign (17th January to 5th February) was heavily impacted by smoke pollution from the bushfires affecting the region. On 5th February, a substantial rain event extinguished the fires and cleared the atmosphere of residual smoke pollution (Mouat et al., 2022; Simmons et al., 2022). The smoke pollution period has been removed from the analysis presented here as we focus on understanding atmospheric processes during more normal conditions. Thus, this paper presents the analysis of BVOCs alongside anthropogenic emissions

and their role in NPF during *the second part of the COALA-2020 ambient measurements campaign running*  116
*from February 5th until March 17th 2020.*  117

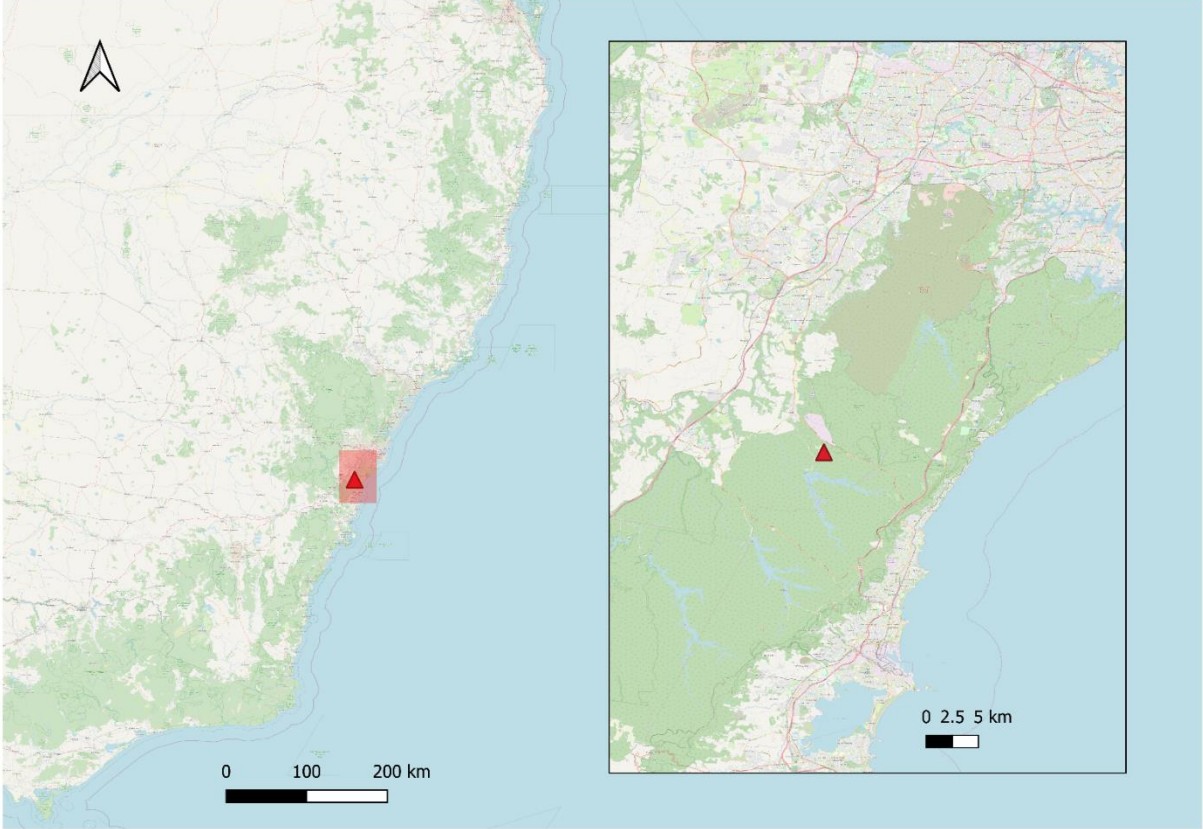

118

*Figure 1 Location of the sampling site, to Sydney, NSW in the north. The sampling site had four different climate control containers for the*  119

*instruments, as well as a soil sampling site around 50 meters northeast from the main sampling site and the High-Vol PM filter. "Map data*  120

*copyrighted OpenStreetMap contributors and available from https://www.openstreetmap.org"*  121

**2.2 Instrumentation**  122

The instruments deployed in the campaign are presented in Table 1. They included an air quality monitoring  123
station owned and operated by the NSW Government Department of Climate Change, Energy, the  124
Environment and Water (DCCEEW**)**, located approximately 10 m away from the main sampling line for VOCs.  125
This station included measurements of temperature, windspeed and direction, $PM_{10}$, $PM_{2.5}$, $O_3$, $SO_2$, $NO_x$, CO  126
and visibility. Inlet heights on this station were between 4.5m to 5.6 m above ground level. All NSW air quality  127
monitoring stations are accredited by the National Association of Testing Authorities (Australia), however it  128
should be noted that these instruments are targeted at regulatory standards and are not research grade. In  129
particular this means that measurements made close to the detection limits are likely to be inaccurate and  130
should be interpreted as indicative measures rather than accurate quantitative measures of atmospheric  131
concentrations.  132

VOCs were measured using a Proton Transfer Reaction Mass Spectrometer (Ionicon PTR-ToF-MS 4000) which  133
operated with a mass range spanning m/z = 18-256. The drift tube was held at a temperature of 70° C,  134
pressure at 2.60 mbar, and an electric field to molecular number density ratio of 120 Td. The instrument was  135
housed in a separate climate-controlled unit. Samples were drawn from an inlet on a 10 m mast through a  136
20 m long PTFE line using a bypass flow of 1.2-3 L $min^{-1}$. Calibrations were made on site using standardized  137

cylinders containing 17 compounds including isoprene, monoterpenes, methyl vinyl ketone (MVK) & methacrolein (MACR), benzene, $C_8$-aromatics, and $C_9$-benzenes (Mouat et al., 2022). Mass spectra were integrated to produce data at 1 minute temporal resolution. Mole fractions were further averaged on a five-minute basis.

A suite of aerosol instruments were operated within in the Atmospheric Integrated Research Facility for Boundaries and Oxidative eXperiment (AIRBOX) container (Chen et al., 2019). Sample air was drawn from a common aerosol bypass inlet. The inlet was located 5 m above ground level for the following instruments:

1. A Ultrafine Condensation Particle Counter (UCPC TSI 3776) was used to measure condensation nuclei number concentration greater than 3 nm ($CN_3$) (TSI Incorporated, Shoreview, MI, USA). The instrument was operated at a sample flow rate of 300 mL min$^{-1}$. Measurements were recorded at 1 Hz temporal resolution.

2. A Scanning Mobility Particle Sizer (SMPS) was used to measure aerosol size distribution between 14 and 670 nm mobility diameter. Full scans of this size range were recorded every five minutes. The system consisted of an X-ray aerosol neutralizer and 3071 Long Electrostatic Classifier (TSI Incorporated, Shoreview, MI, USA) coupled to a 3772 CPC (TSI Incorporated, Shoreview, MI, USA). Sample was drawn from the same inlet as used by the UCPC.

3. Chemical composition of aerosols with diameter smaller than 1 μm ($PM_1$) were taken using a Time-of-Flight Aerosol Chemical Speciation Monitor (ACSM; Aerodyne Research Inc., Billerica, MA, USA). Mass concentrations of organics (Org), sulphate ($SO_4^{2-}$), nitrate ($NO_3^-$), ammonium ($NH_4^+$), and chloride ($Cl^-$) in the aerosol fraction 40-1000 nm vacuum aerodynamic diameter range, referred to as $PM_1$, are reported. Measurements were taken at 10-minute resolution. Sample air was drawn from the aerosol inlet common to the CPC and SMPS and dried using a Nafion dryer to < 40% relative humidity before sampling.

*Table 1: Instruments deployed during the COALA 2020 campaign and included in the present analysis.*

| Name of parameter | Instrument type |
|---|---|
| $NO$ - $NO_2$ - $NO_x$ | API T204 |
| $O_3$ | Ecotech 9810 |
| $PM_{10}$ | Thermo (TEOM) 1405A |
| $PM_{2.5}$ | Thermo (BAM)5014i |
| $SO_2$ | API T100 |
| Black Carbon | Magee Scientific Aethalometer AE33 |
| VOCs | PTR-ToF-MS (Ionicon) |
| $CO$ - $CO_2$ - $CH_4$ - $N_2O$ | FTIR in situ analyser |
| $CN_3$ | TSI 3776 |
| Particle number size distribution (14 nm to 660 nm) | SMPS |
| $PM_1$ mass composition | Tof-ACSM, Aerodyne |
| Wind Speed and Wind Direction | 2D Ultrasonic anemometer |
| Temperature, Relative humidity | Vaisala HMP155 |
| Photosynthetic active radiation (PAR) | |

## 2.3 NPF Classification Method

The method proposed by Dal Maso et al. (2005) was used to classify the particle size distribution data. To apply the method the particle number density plots were made for each day during the campaign and the plots were visually inspected to identify if an event occurred on that day. A day of data was classified as an *event* if there was nucleation, and growth up to 25nm for at least two hours.

Once the events were classified, a logarithmic fit was applied to determine the geometric diameter of each mode. The data was manually divided in chunks of 10 minutes to visually inspect and determine the number of modes and the geometrical diameter range of each event (nucleation <25 nm, Aitken 25 nm -100 nm, accumulation >100 nm). Once those parameters were defined and included in the code, each event was divided in periods of time with similar distribution modes.

For illustration a hypothetical event lasting two hours was divided in two: one hour with simultaneous two particle modes (nucleation and Aitken) and then one hour with just one particle mode (Aitken). This is done to estimate an accurate geometrical particle diameter based on the number of modes. This avoided the problems of changes in the number of modes in the sample. Finally, the data was merged again to have a time series of number of particles predicted with the fit, number of modes predicted and geometrical particle diameter.

The algorithm works by providing the number of modes observed in the input dataset. Then it selects the provided model equation for each mode number and iterates over a hundred fits looking for the best fit. The Bayesian Information Criterion (BIC) was used to identify the best fit, looking for the lowest values. Once the best fit was selected, the total particle number estimated by the model was compared with the sample record for each sample to assure it was within a 5 % difference compared to the total particle number reported in the sample. The result was then visually checked looking for the geometrical diameter and how it compares to the distribution size plots from the raw aerosol distribution size data. Once the model was considered representative and accurate enough, the growth rate for each event was determined using a simple linear regression of the change in the geometrical diameter in time from nucleation to Aitken and eventually to accumulation mode.

## 3. Results and Discussion

### 3.1 Frequency of NPF Events

Of the 40 days included in the analysis, nine days didn't have any data. Of the 31 days with data,  12 (39%) showed clear NPF events, nine (29%) were considered undefined and ten (32%) didn't have enough data or were classified as a non-event  The percentage of days with NPF is similar to those of other sites in forested areas in the Northern Hemisphere (Kalkavouras et al., 2020; Uusitalo et al., 2021). 39% of days with NPF events and 29% with undefined events implies a highly reactive atmosphere even in this rural area with some anthropogenic influence of mobile sources and occasionally coal-fired power plant in the Hunter Valley region.

Figure 2 illustrates the time series of an NPF event observed on $11^{th}$ February 2020. The NPF event commenced at 8 am, preceded by a peak in both $SO_2$ concentrations and the estimated $H_2SO_4$ proxy. The shaded area in the plot highlights the growth period which is marked by an increase in mode diameter and

condensation sink. Ozone is also increasing at this time. The increase in aerosol $SO_4^{2-}$ and organics during this period shows the influence of this reaction chemistry on particles larger than 100 nm in the aerosol size distribution. We estimated the $H_2SO_4$ proxy using the rural model developed by Dada et al (2020). This model was chosen from among the options because the environmental conditions under which it was derived are the most similar to those of our sampling site. The equation used to estimate $H_2SO_4$ proxy was

$$[H_2SO_4]_{\{rural\}=} - \frac{CS}{2*(2x10^{-9})} + [\left(\frac{CS}{2*(2x10^{-9})}\right)^2 + \frac{[SO_2]}{(2x10^{-9})} * (9 * 10^{-9} * GlobRad]^{\frac{1}{2}},$$

Where CS is the condensation sink, $SO_2$ is the concentration of $SO_2$, GlobRad is the global radiation obtained from the Photosynthetic Photon Flux Density (PPFD) values as GlobRad = 0.327*PPFD.

210

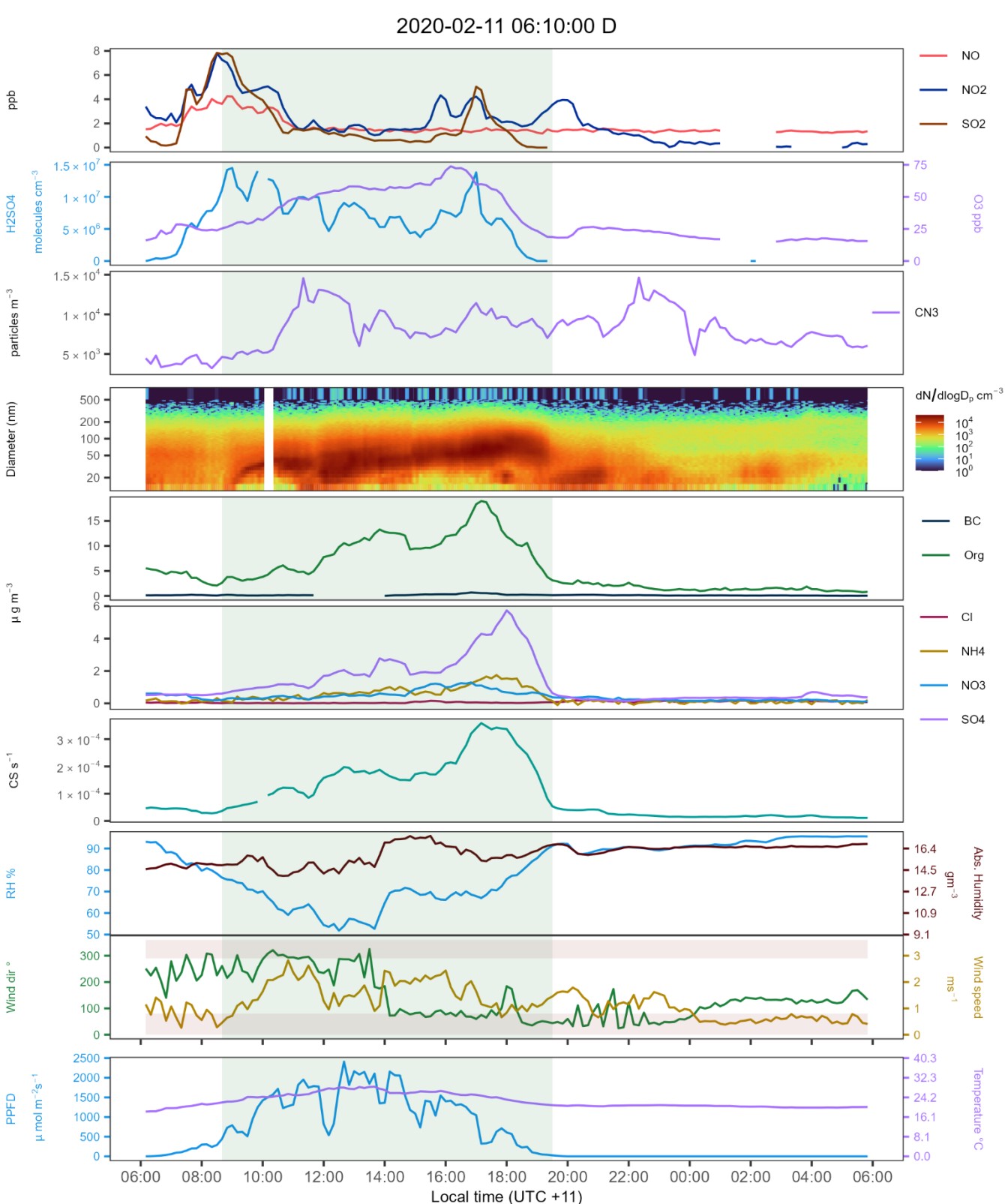

Figure 2 Time series for all selected variables during the NPF event during 2020-02-11. NO = Nitric oxide, NO₂ = Nitrogen dioxide, SO₂ =

Sulphur dioxide, H₂SO₄ = Sulphuric acid, O₃ = Ozone, CN₃ = Condensation Nuclei >3nm, CN₃-CN₁₄ = difference of CN₃ minus the sum of all

channels from the SMPS data. BC = Black carbon. Org = Organic mass fraction, NH₄ = Ammonium mass fraction, NO₃ = Nitrates mass fraction,

211
212
213
214

*SO$_4^{2-}$ = Sulphates mass fraction, Cl = Chloride mass fraction. CS = condensation sink. PFFD = Photosynthetic Photon Flux Density. VOCs mole* 215
*fractions were not available during this specific event. Note how the fraction of organics, sulphates and ammonium increase with a positive* 216
*correlation, dominating over the nitrate and chloride fractions until the end of the event. The light green area marks the NPF, and growth period* 217
*mentioned in the analysis. the brown shade areas in the wind panel highlight areas where the wind comes from the nearby roads. Note that the* 218
*NO values are close to detection limit and look biased high and hence should be interpreted as an indicative rather than accurate quantitative* 219
*measure of atmospheric concentration.* 220

## 3.2 Triggers for NPF Events 221

Of the twelve days with NPF, four occurred during the night or early morning (before sunrise), and eight 222
during the day. The time series of SO$_2$, NO$_x$, ozone, VOCs and the aerosol composition were used to identify 223
which variables triggered and influenced the aerosol formation and growth. Of the twelve event days of NPF, 224
six days include VOC data and eight days include aerosol composition data, noting that the composition data 225
is not applicable to particles <100 nm and only three events led to accumulation sized particles (diameter 226
>100 nm). The data available for each event is summarised in Table 2. 227

*Table 2 Data available for each NPF event identified during the COALA campaign* 228

| Event | time | NOx | O₃ | SO₂ | VOCs | CN₃ | SMPS | ACSM |
|---|---|---|---|---|---|---|---|---|
| 05/02/2020 | N | ✓ | ✓ | ✓ | ✓ | ✓ | ✓ | |
| 10/02/2020 | N | ✓ | ✓ | ✓ | | ✓ | ✓ | ✓ |
| 11/02/2020 | D | ✓ | ✓ | ✓ | | ✓ | ✓ | ✓ |
| 15/02/2020 | D | ✓ | ✓ | ✓ | | ✓ | ✓ | ✓ |
| 16/02/2020 | D | ✓ | ✓ | ✓ | | ✓ | ✓ | ✓ |
| 24/02/2020 | N | ✓ | ✓ | ✓ | | ✓ | ✓ | ✓ |
| 06/03/2020 | D | ✓ | ✓ | ✓ | ✓ | ✓ | ✓ | ✓ |
| 07/03/2020 | D | ✓ | ✓ | ✓ | ✓ | ✓ | ✓ | ✓ |
| 08/03/2020 | D | ✓ | ✓ | ✓ | ✓ | | ✓ | ✓ |
| 09/03/2020 | N | ✓ | ✓ | ✓ | | ✓ | ✓ | |
| 10/03/2020 | D | ✓ | ✓ | ✓ | ✓ | ✓ | ✓ | |
| 11/03/2020 | D | ✓ | ✓ | ✓ | ✓ | | ✓ | |

229

From the daily time series of all available variables over the 12 days of NPF events, it is evident that SO$_2$ 230
frequently triggers or at least influences the particle formation. However, the trigger for nighttime events is 231
pointing to NO$_2$ related chemistry but without complementing measurements it's unclear. To group the 232
common factors influencing NPFs for daytime and night-time events, a comparison of the growth rate was 233
used to determine whether the rates were similar during the day and during the night. 234

235

236

### 3.3 Particle Growth Rates during daytime and nighttime events

The estimated growth rate is presented in Figure 3. Only four of the nine events during daytime (upper panel of Figure 3) had a representative Pearson coefficient ($R>0.6$), the remaining five events did not have a stable linear growth and are not shown in the plot. The events which showed unstable growth patterns suggests a highly variable condensation source, possibly resulting from changing $H_2SO_4$ concentrations. This is complicated further by changing wind directions.

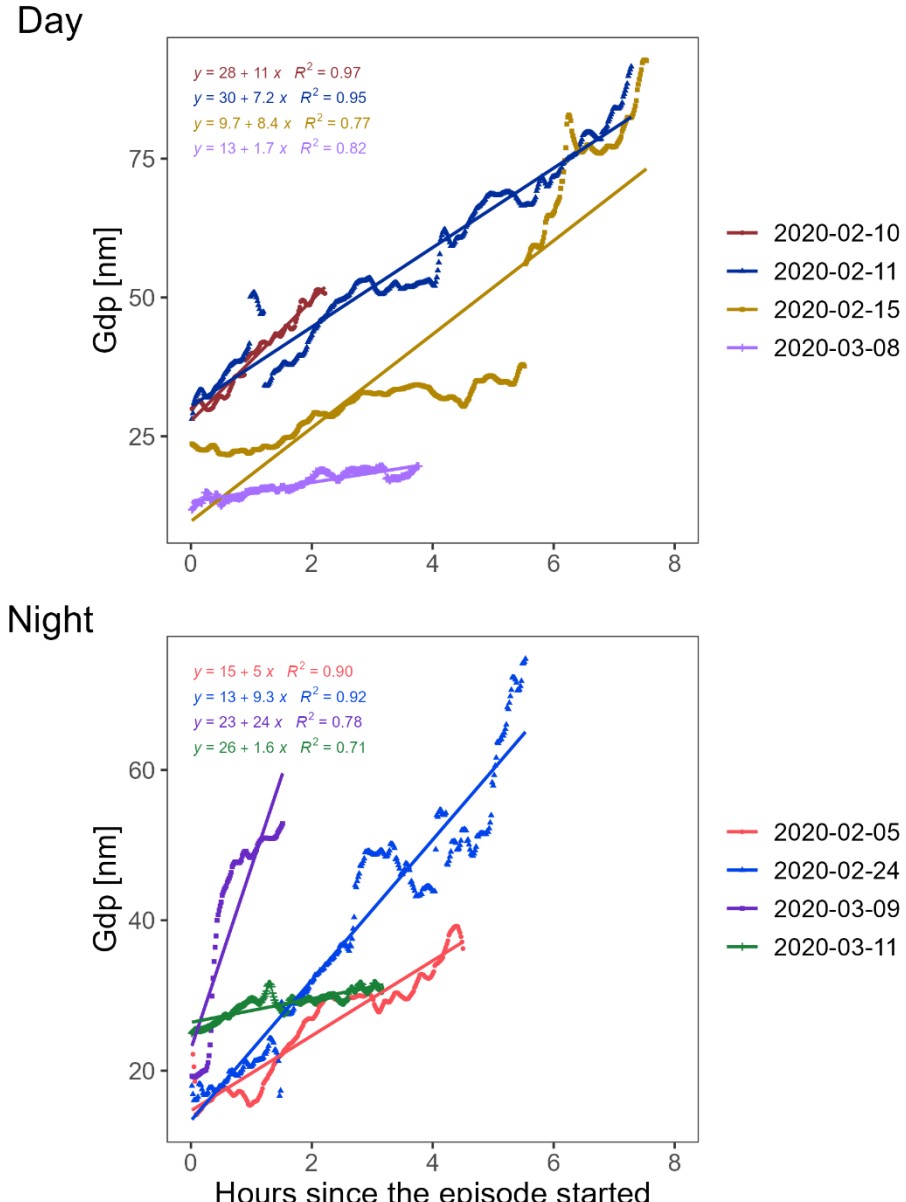

Figure 3: Geometric particle diameter evolution in each event where the logarithmic fit converged. The top panel presents the daytime data (only four events converged to a statistically significant model). The bottom panel presents the nighttime events.

Some events highlight how the dynamic nature of daytime concentrations of $O_2$, $NO_2$ and $O_3$ complicate the analysis (see figure S1). Nonetheless, these events provide insight into the factors that may drive the growth and particle formation and so were included in all the analysis. Event 2020-02-15 in Figure 3 is an example

of how the geometric particle diameter can change when there is rapid growth. The first part of the 249
regression shows slower growth rate. After the 6th hour of slow growth, the rate increases substantially, 250
attributed to an increase of $H_2SO_4$ around this time. Shortly after this accelerated growth, there is a wind 251
change from northerly to southerly (Figure S4). Following the southerly wind shift, a lower condensation sink 252
and higher relative humidity likely contributed to the Gdp increase via enhanced condensation and water 253
uptake. Declining tracer levels $SO_2$ and $NO_x$ indicate that local particle growth mechanisms were likely 254
dominant over the influence of a new air mass up to the 7th hour when increases in $NO_x$ and $SO_2$ are observed. 255

In contrast to the daytime events, all the night-time events were stable enough to determine the event 256
growth rate. The growth rate varied considerably between events (see lower panel of Figure 3) and most 257
likely reflects differences in the factors driving the particle formation between these episodes. The specific 258
oxidation pathways that were active during each event likely had a direct impact on the observed differences 259
in growth rates. These reaction pathways might include monoterpene ozonolysis and condensation over 260
previously formed clusters (Liu et al., 2023; Wang et al., 2023), or oxygenated VOCs (OVOCs) brought to the 261
site and condensed on formed seeds or may initiate nucleation (Bianchi et al., 2019; Higgins et al., 2022). 262
Some of these processes were observed during the campaign and will be further explored on the nighttime 263
events section. 264

**3.4 Daytime NPF Events** 265

From the timeseries analysis of all daytime events (see Figure 2, 5-6 and supplementary figures S1-S4), four 266
key points were identified for NPF in the area: 267

1. $SO_2$ arriving at the site appears to trigger nucleation and growth events. 268
2. VOC availability (monoterpenes and isoprene) enhances nucleation and growth. 269
3. The hours with high VOCs concentrations and higher oxidation capacity in the atmosphere ($OH$ 270
   concentrations are assumed to be higher during the hours with higher PAR) have higher particle 271
   number concentrations and generally guaranteed growth up to the accumulation mode. 272
4. Growth without the influence of $SO_2$ may occur but will do so at a slower rate. 273

During most of the daytime events $SO_2$ and $NO_2$ plumes impacted the site at some stage of each event. 274

On some occasions the $SO_2$ plume might last for a couple hours as shown in the first part of the event on Feb 275
11th 2020 (see Figure 2), whilst at other times there were multiple peaks of high $SO_2$ measured at the site as 276
shown in several other events in the record (e.g. Figures S2, S3, S4). However, subsequent nucleation was 277
observed on every occasion that $SO_2$ was observed above the detection limit at the site, growth occurred 278
within 0 to 150 minutes after the $SO_2$ was first detected. The time window difference between events reflects 279
the influence of conditions at the start of a particle growth event. To highlight this phenomenon a cross 280
correlation between $SO_2$ and the aerosol mass of aerosol $SO_4^{2-}$ time series obtained from the tof-ACSM and 281
the measured particle number concentration ($CN_3$) was applied. Figure 4 shows the Pearson correlation 282
between $SO_2$ and the $CN_3$ and aerosol $SO_4^{2-}$ in a window period of four hours i.e. starting two hours before 283
the nucleation commenced and ending after the first two hours of the event. This time window captures the 284
$SO_2$ influence on the particle formation. Each line/point shows the correlations at 0, 30, 60, 90, 120 and 150 285

minutes lagged for each daytime event. The dotted blue lines show where the lagged correlation is significant at ($|r| > 0.5$).

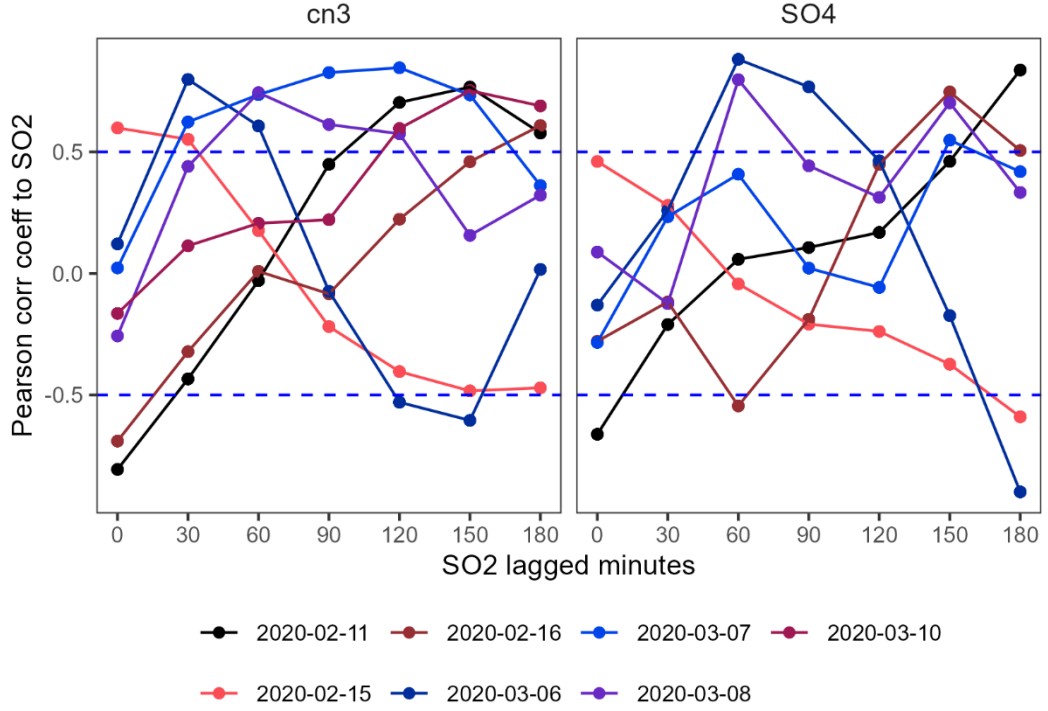

*Figure 4 : Pearson correlation values obtained from the cross correlation between $SO_2$ and $CN_3$ and $SO_4^{2-}$ mass. The dashed lines represent the 0.5 threshold as a reference to identify significant correlations. Events on Feb 10th and March 11th did not follow this pattern and were removed from the plot.*

To interpret Figure 4, we can use the event on February 11th (black line) as an example. Here the correlation between $SO_2$ and $CN_3$ becomes significant (at $|r| > 0.5$) if the $SO_2$ time series is lagged 120 minutes with respect to the aerosol data; and the correlation between $SO_2$ and aerosol $SO_4^{2-}$ becomes significant after 3 hours. This means that if we move the $SO_2$ time series two hours forward it will be better correlated with the particle number concentration, accounting for the reaction time of $SO_2$ to produce $H_2SO_4$ and enhance/trigger the particle formation under the conditions in the atmosphere at the time. Usually, the $SO_2$ correlation with aerosol $SO_4^{2-}$ needs a longer lag time to be significant. This is a potential indication of the order in which the chemical reactions happen. First, we will see oxidation of the $SO_2$ to $H_2SO_4$ then nucleation, and finally growth in mass when there is condensation or coagulation near CCN sizes. Using time series analysis as shown here can provide more evidence when the chemical mechanisms are known but observations of other variables are not available.

A similar result is observed for other events at different lagged times. The difference in the time necessary to achieve a significant correlation between $SO_2$ and the particle number seems to be related to the quantity of VOCs available when the $SO_2$ plume arrives at the site. This aligns with our understanding of the transition from nucleation to particle growth. In the early hours, observed monoterpene levels are sufficient to drive nucleation through ozonolysis and subsequent HOM formation (Iyer et al., 2021; Kirkby et al., 2023; Wang

et al., 2023). Particle growth was observed later in the day (see Figure 2 for example) likely driven by the condensation of OVOCs. The increase in the sulfate fraction observed in the ACSM supports the condensation of sulfate-related species onto the growing particles. Events on February 15$^{th}$, March 6$^{th}$, and March 7$^{th}$ showed the highest correlations within the first 30 minutes of lagging the data. Common to these events were relatively high levels of monoterpenes (~1 ppb either directly observed or inferred from high PAR and temperature) in the hour before NPF detection at the site (see Figure S1, S2, and S4). The elevated monoterpene levels and subsequent ozonolysis likely initiated particle formation during these times, with available $H_2SO_4$ further facilitating nucleation. The HOM proxy (monoterpenes*ozone (e.g.: Zhang et al., 2024)) also peaked during this period, supporting the idea that HOM formation via ozonolysis was a dominant oxidation pathway driving initial nucleation.

The event on March 8$^{th}$ also met this condition (see Figure 5), although it exhibited a relatively low growth rate. Elevated isoprene and MACR + MVK concentrations during this event suggest the potential for isoprene to suppress new particle formation, as described by Heinritzi et al. (2020). Higher isoprene levels after 12:00, accompanied by increased MACR + MVK coincided with a decline in the number of smaller particles (although $CN_3$ data is incomplete). This is the first step in the reaction chain to produce $C_{15}$ dimers. This observation aligns with the HOM proxy (monoterpenes*ozone): higher proxy values corresponded to periods of higher particle numbers, while a decrease in the HOM proxy coincided with a decrease in particle numbers and an increase in MACR + MVK products, suggesting a shift towards more isoprene-influenced atmospheric chemistry. Concurrently, increases in the organic and sulfate fractions, along with the condensation sink, indicate a shift towards conditions favoring the growth of existing larger particles through condensation and coagulation, rather than nucleation events.

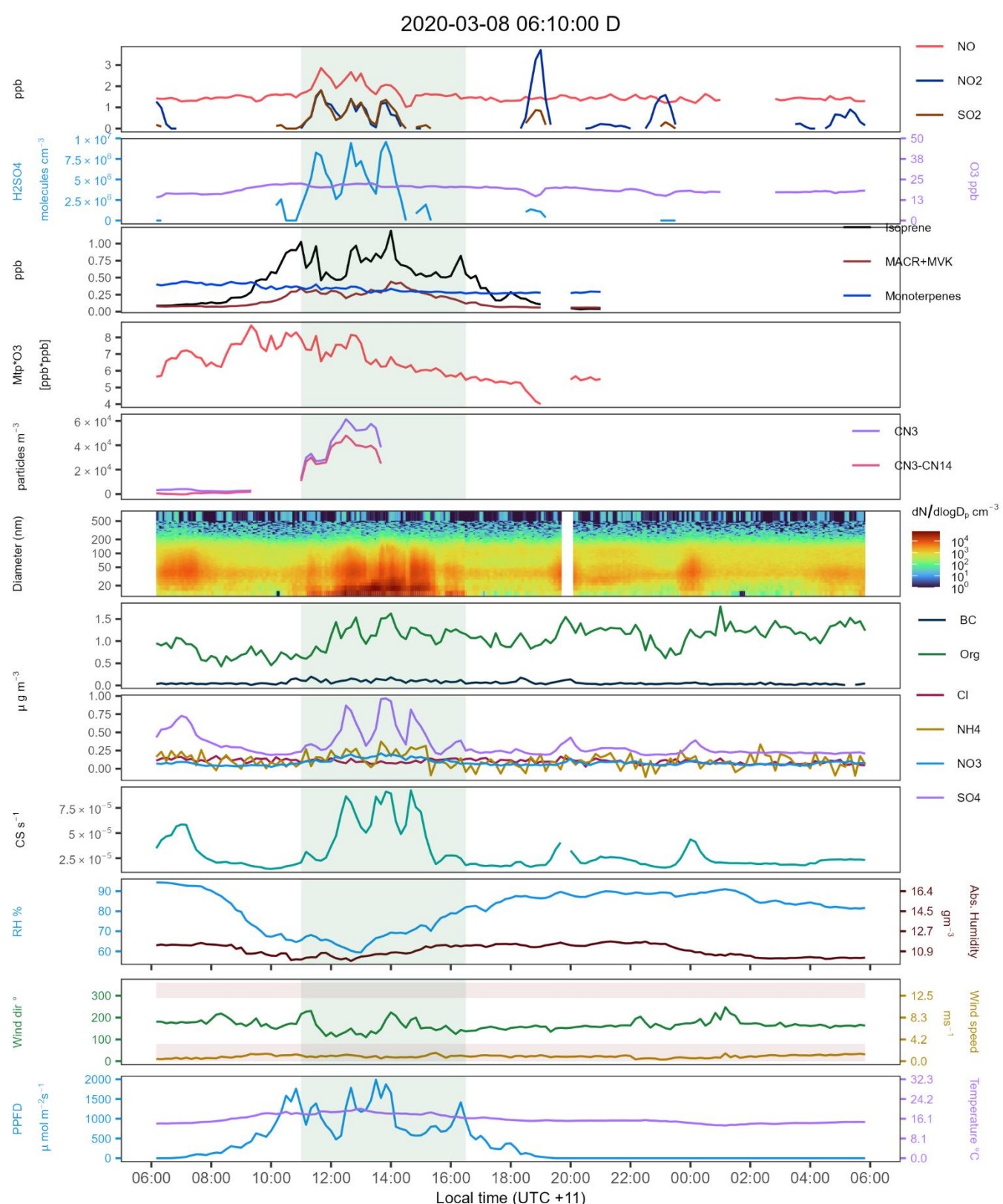

Figure 5 Time series of all selected variables during the NPF event during 2020-03-08. NO = Nitric oxide, NO₂ = Nitrogen dioxide, SO₂ = Sulphur dioxide, H₂SO₄ = Sulphuric acid, O₃ = Ozone, MACR+MVK = isoprene ox. products methacrolein and methyl-vinyl-ketone, CN₃ = Condensation

329

330

331

*Nuclei >3nm, $CN_3$-$CN_{14}$ = difference of $CN_3$ minus the sum of all channels from the SMPS data. BC = Black carbon. Org = Organic mass fraction, $NH_4$ = Ammonium mass fraction, $NO_3$ = Nitrates mass fraction, $SO_4^{2-}$ = Sulphates mass fraction, Cl = Chloride mass fraction. CS = condensation sink. Mtp*ozone = HOM proxy product monoterpenes and ozone [ppb*ppb].*

The Feb 11$^{th}$ and Feb 16$^{th}$ events had similar arrival times for the $SO_2$ pollution (8:00 to 9:00) although the photochemistry was not fully active yet (see $H_2SO_4$), monoterpenes levels were consistently high during all the campaign (~0.4 ppb based on the days with data), enough to promote nucleation.  This presumption is supported by looking at the event on February 16$^{th}$ (see Figure S3). In this event, a first peak of $SO_2$ at 8:00 started nucleation but then condensation or coagulation dominated favouring growth. The $CN_3$ -$CN_{14}$ data show that after that initial nucleation period the particle number is dominated by the >14nm fraction. Multiple $SO_2$ plumes reaching the site produced higher ratios of $H_2SO_4$, but promoted growth to larger particles sizes particularly on the sulphates fraction that correlates with the $SO_2$ peaks. In the evening there were a couple of small particle bursts that were quickly coagulated on larger size particles.

On March 10$^{th}$ (see Figure 6), a sharp decline in high monoterpene concentrations was observed just before the aerosol event. The aerosol growth phase is then observed to correlate with peaks in $SO_2$ and $NO_X$, as well as elevated levels of isoprene. This suggests monoterpene ozonolysis initiated nucleation, and the observed particle growth coincided with periods indicative of increased atmospheric pollution, potentially contributing condensable material.

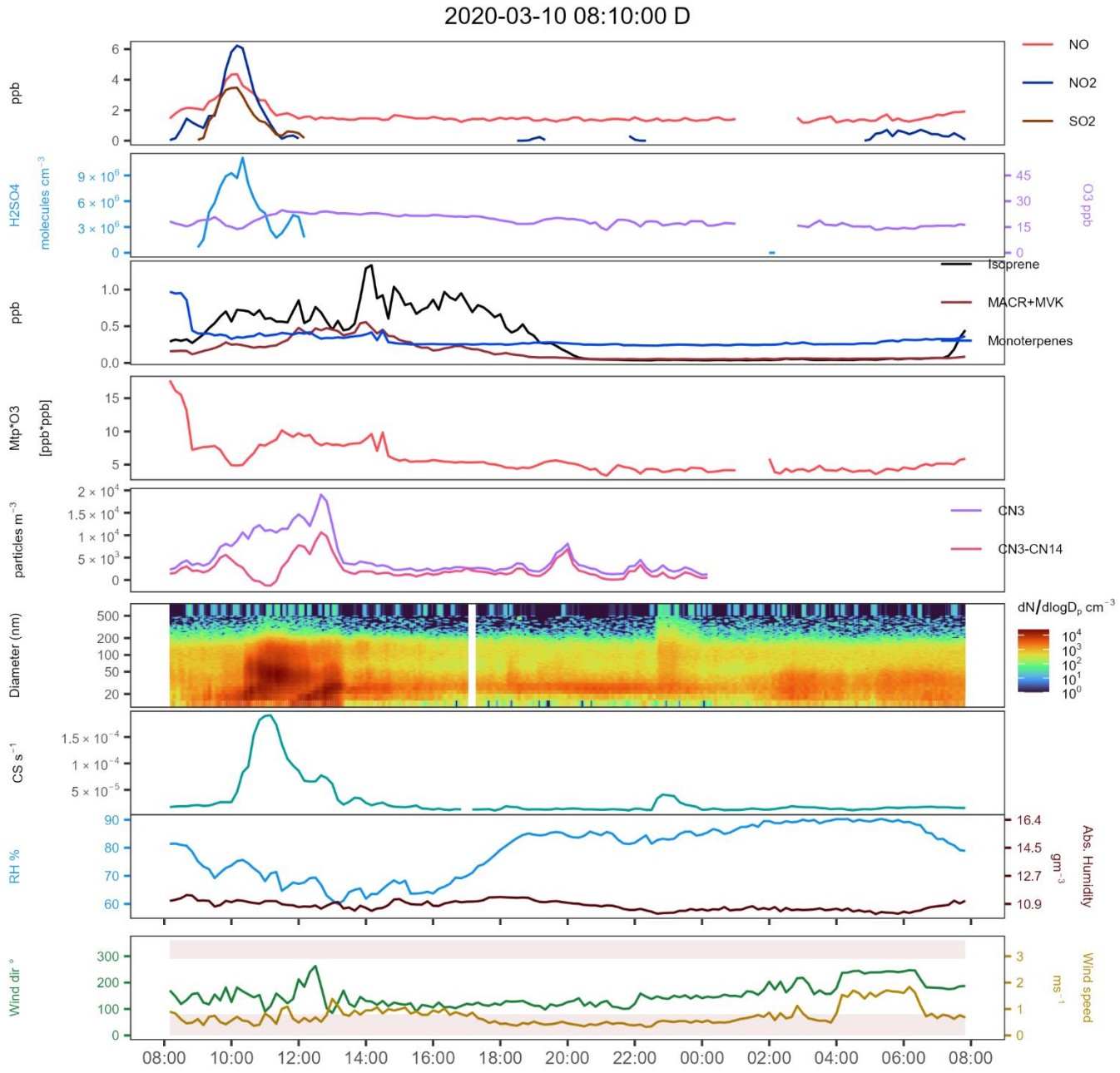

2020-03-10 08:10:00 D

Figure 6 Time series of all selected variables during the NPF event during 2020-03-10. The drop of $CN_3$ seem related to the lack of $SO_2$ after 11:00. NO = Nitric oxide, $NO_2$ = Nitrogen dioxide, $SO_2$ = Sulphur dioxide, $H_2SO_4$ = Sulphuric acid, $O_3$ = Ozone, MACR+MVK = isoprene ox. products methacrolein and methyl-vinyl-ketone, $CN_3$ = Condensation Nuclei >3nm, $CN_3$-$CN_{14}$ = difference of $CN_3$ minus the sum of all channels from the SMPS data.. CS = condensation sink. Mtp*ozone = HOM proxy product monoterpenes and ozone [ppb*ppb].

For all daytime events $SO_2$ and $NO_2$ are significantly correlated with a Pearson correlation of 0.78, suggesting a common source for both pollutants. The closest source of combustion products is the Appin Road located north of the sampling site. Given that the sampling site is away from other possible sources of $SO_2$ and $NO_2$ and the relatively low wind speeds during most of the campaign (see Figure S6), combustion from mobile

sources is considered the most likely source of both compounds but there might be some influence of more distant coal-fired power stations. Another factor to contribute to this theory is that the $SO_2$ levels were higher during the day when most of the commuting takes place and leading to a higher vehicle density on the roads. The intermittent $SO_2$ and $NO_2$ peaks suggest the influence of mobile sources with poor emission control onboard. The effects of vehicles with poor emission control technologies on ambient concentrations of $SO_2$, $NO_x$, AVOCs and PM has been seen in different studies (Kari et al., 2019; Phillips et al., 2019; Smit et al., 2019) and the legislation controlling fuel standards and emissions is relatively lax in New South Wales (Paton-Walsh et al., 2019).

During the COALA-2020 campaign, many events, such as the one on February 16[th] (Figure S3), exhibited elevated gas-phase SO2. The availability of monoterpene to form highly condensable ULVOC/ELVOC is crucial in the observed events. While the oxidation products of isoprene can also condense on pre-existing particles (Stangl et al., 2019), the dominant pathways and their efficiency are likely driven by monoterpenes. Although VOC data was not available for February 16[th], the consistent diurnal profile of VOCs observed throughout the remaining dataset (Figure S5) suggests enhanced monoterpene and isoprene availability during the daytime. Under these conditions of available BVOCs, particle growth was frequently observed, suggesting a contribution from condensed organic material. As the night approaches and BVOC emissions decrease with temperature, the remaining OVOCs can undergo further oxidation, forming less volatile species that are more prone to condensation on existing particles. However, the limited availability of VOCs after their consumption (estimated around 22:00 based on diurnal cycles in Figure S5) likely limits further growth.

When there is negligible $SO_2$ in the atmosphere but high VOC concentrations (particularly monoterpenes), autooxidation processes can be initiated, potentially leading to both nucleation and subsequent particle growth (Bianchi et al., 2019). Growth was observed during the first event on February 10[th] (see daytime data in Figure 7) despite low $SO_2$ and may be related to the condensation of HOMs formed through monoterpene autooxidation. The average concentration of monoterpenes during the campaign in the morning was often sufficient to initiate reactions leading to ULVOC that favor both new particle formation and the growth of pre-existing particles.

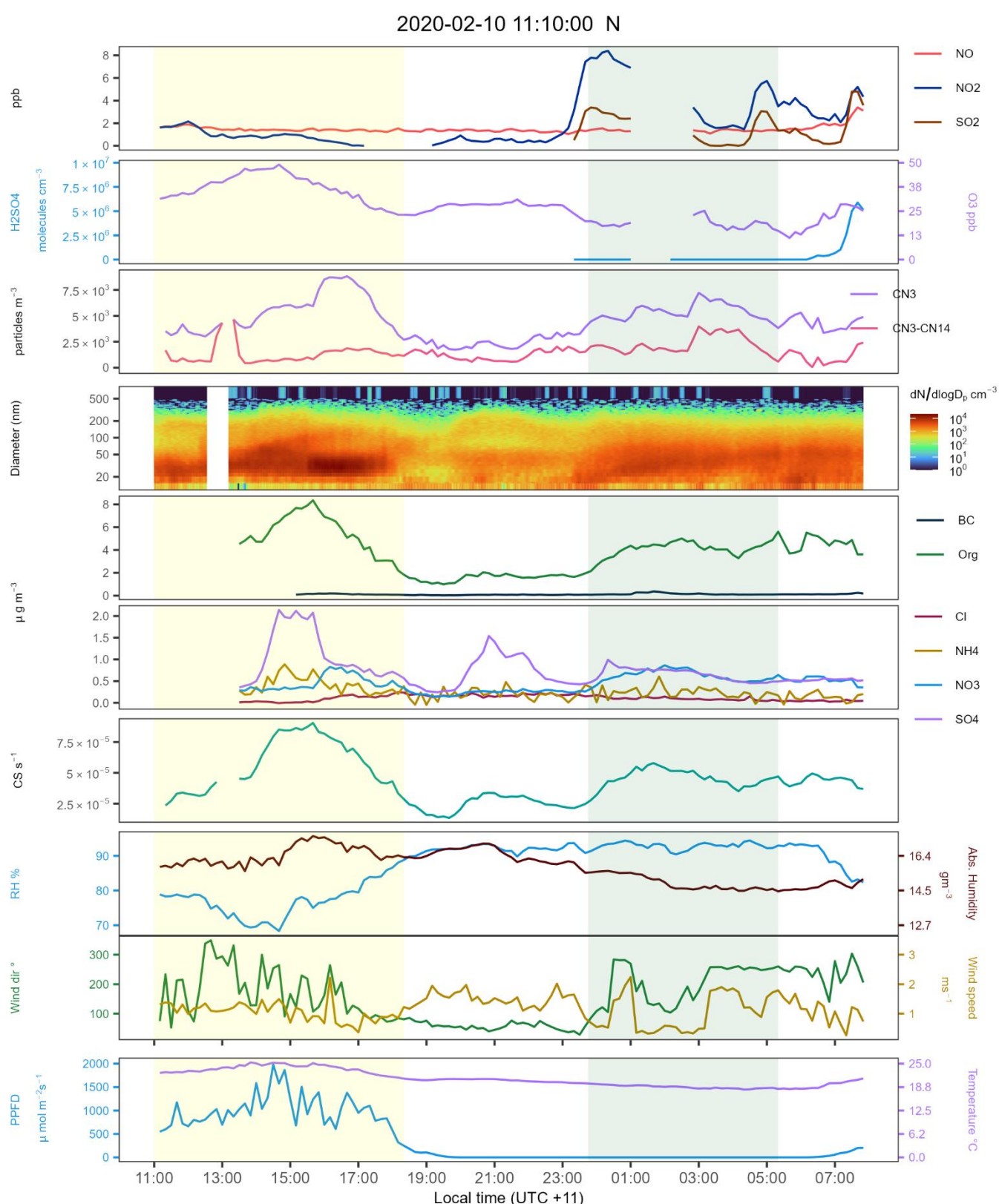

*Figure 7: Time series for all selected variables during the NPF event during 2020-02-10. NO = Nitric oxide, NO₂ = Nitrogen dioxide, SO₂ =*

*Sulphur dioxide, H₂SO₄ = Sulphuric acid, O₃ = Ozone, CN₃ = Condensation Nuclei >3nm, CN₃-CN₁₄ = difference of CN₃ minus the sum of all*

386

387

388

*channels from the SMPS data. BC = Black carbon. Org = Organic mass fraction, NH$_4$ = Ammonium mass fraction, NO$_3$ = Nitrates mass*   389

*fraction, SO$_4{}^{2-}$ = Sulphates mass fraction, Cl = Chloride mass fraction. CS = condensation sink. Mtp*ozone = HOM proxy product monoterpenes*   390

*and ozone [ppb*ppb]. VOCs mole fractions were not available during this specific event. Note how there does not seem to be any significant SO$_2$*   391

*or NO$_2$ pollution prior to the NPF start. At the same time of the particle growth there are enhancements in the organic, sulphate and ammonium*   392

*mass fraction. There are two events in this plot. One in the morning with an unknown start and ending around 18:00, and the other at night.*   393

*The light green area marks the night event, and the yellow highlight refers to the daytime event.*   394

Australia experiences an isoprene-dominated atmosphere (Emmerson et al., 2016; Ramirez-Gamboa et al., 2021), and the chemical balance in the atmosphere can rapidly change, particularly in the hotter seasons when more isoprene is emitted. While SOA formation on pre-existing particles can involve molecules with relatively high saturation vapor pressures, new particle formation critically depends on molecules with extremely low saturation vapor pressures due to the Kelvin effect (Tröstl et al., 2016). Heinritzi et al. (2020) showed that reducing $C_{20}$ formation (α-pinene oxidation in isoprene presence) to favor $C_{15}$ formation reduces nucleation rates. However, it is also important to highlight that $C_{15}$, $C_{10}$, and even $C_5$ oxidation products from isoprene oxidation can contribute to SOA mass on existing particles. Therefore, in Australia's isoprene-dominated environment, higher isoprene to monoterpene ratios could lead to a greater production of $C_5$ and $C_{15}$ products that contribute to particle growth on existing aerosols (and SOA mass), while simultaneously hindering new particle formation by reducing the formation of $C_{20}$ dimers from monoterpenes.   395–406

## 3.5 Night-time NPF Events   407

We observed three nighttime events during COALA. Unfortunately, none of these events coincided with all data sets being collected which limits our ability to discuss the reactions driving the nighttime events. Consistent between all nighttime events is an increase in particles ($CN_3$), elevated $NO_2$, and an increasing condensation sink. Unfortunately, the $NO_x$ instrument available in this study was not ideal for this type of measurement for several reasons: it is not designed to be sensitive to the low $NO_x$ levels observed in rural areas; it is not capable of separating $NO_x$ from $NO_y$; and it was set up to calibrate in the night hours between 1:00 and 2:00 every day. Nonetheless, during the night-time events the particle size distribution data and the $CN_3$ data showed particle formation and growth from nucleation to Aitken modes when there were considerable increases of $NO_2$ and simultaneous decreases in ozone.   408–416

When VOC data are available, monoterpene concentrations were moderate and increased steadily during the event (5[th] Feb and 9[th] March). Isoprene was high at the start of the event on 5[th] Feb, (see Figure 8) however the sudden decrease in isoprene concentration likely coincides with sunset on that day. When aerosol composition data was available (10[th] Feb) aerosol organic, nitrate and sulphate concentrations increase during the event. When ozone data were available, concentrations decreased slightly during the course of the event.   417–422

The frequency of nocturnal events observed in this study is lower than observed previously at a nearby location (Tumbarumba a eucalypt forest site located 300 km to the SE of Cataract (Suni et al., 2009)), where in the summer of 2006, nocturnal NPF events were observed on 32% of the analysed nights and occurred 2.5 times more frequently than daytime events. Simulating the NPF at Tumbarumba, Ortega et al (2012) was able to reproduce the observations from Tumbarumba by ozonolysis of 13 -carene to initiate nucleation   423–427

and a-pinene to grow particle diameters. Ozonolysis of limonene was found to contribute to both nucleation and aerosol growth. The lower frequency observed in our study may be linked to the apparent initiations of nucleation by $NO_2$, which nocturnally can react with $O_3$ to form nitrate radicals. Li et al. (2024), suggest even trace amounts of $NO_3$ radicals suppress the NPF.

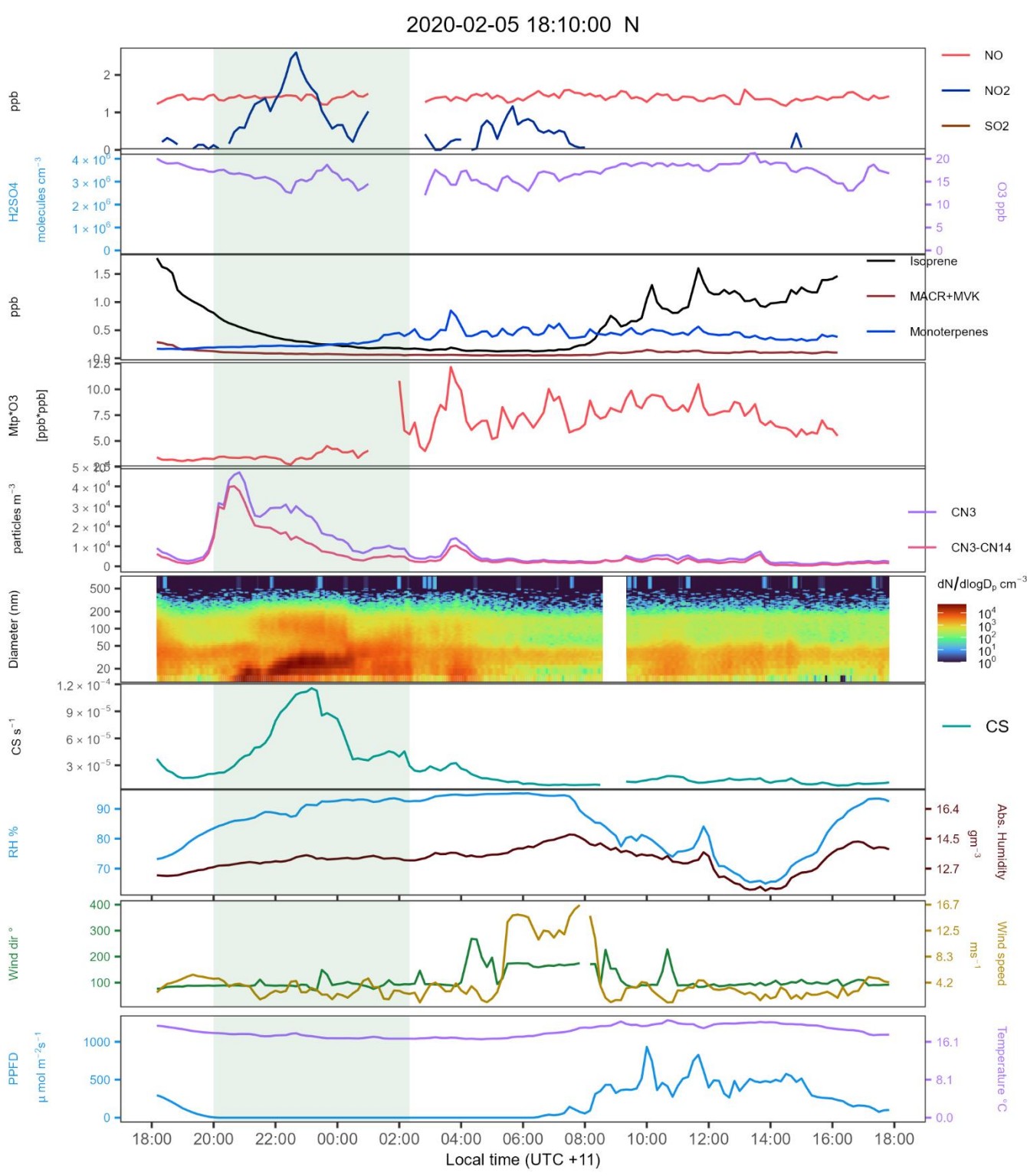

2020-02-05 18:10:00  N

Figure 8: Time series for all selected variables during the NPF event during 2020-02-05. NO = Nitric oxide, NO₂ = Nitrogen dioxide, SO₂ = Sulphur dioxide, H₂SO₄ = Sulphuric acid, O₃ = Ozone, MACR+MVK = isoprene ox. products methacrolein and methyl-vinyl-ketone, CN₃ =

433

434

435

*Condensation Nuclei >3nm, CN$_3$-CN$_{14}$ = difference of CN$_3$ minus the sum of all channels from the SMPS data. BC = Black carbon. Org =*
*Organic mass fraction, NH$_4$ = Ammonium mass fraction, NO$_3$ = Nitrates mass fraction, SO$_4$ = Sulphates mass fraction, Cl = Chloride mass*
*fraction. CS = condensation sink. Mtp\*ozone = HOM proxy product monoterpenes and ozone [ppb\*ppb]. Note how the particle number goes*
*below 10000 after the growth reached Aitken mode (0:00). There is not a substantial increase in the aerosol mass when the particle number and*
*geometrical particle diameter increase. The light green area marks the NPF and growth period mentioned in the analysis.*

## 3.6    Aerosol fraction: Day vs Night

Figure 9 shows the mass fraction of the PM$_1$ aerosol mass measured in the ACSM. Most of the daytime events show a similar mass fraction distribution. The organic fraction is the largest mass fraction followed by sulphates, ammonium, nitrates, and chlorides. We observed higher sulphate mass fractions in days with higher $SO_2$ availability such as the events on Feb 16[th] and March 8[th], where the average sulphate mass fraction was larger or similar to the organic fraction (see Figure 9). These two events also display the highest proportion of ammonium during daytime events. The overall mass during night-time is much lower than during daytime, likely related to the lower concentrations of VOCs available during the night, resulting in growth not reaching sizes where it was detectable by the ACSM. Even with less total mass during the night, the contribution of each fraction is similar to the daytime events. The most notable difference between the mass fractions during day and nighttime NPF events is the higher fraction of chlorides during night-time. Chloride is a primarily sourced aerosol component, so is not influenced by the aerosol formation capacity of the atmosphere at night that reduces the total organic, sulphate, nitrate and ammonia mass but does not impact chlorides.

Something to highlight is the higher fraction of ammonium compared to nitrates through most of the events. Regions with low $NO_x$ have been previously characterized with higher ammonium fractions compared to nitrates (Du et al., 2015; Liu et al., 2022; Petit et al., 2015; Takami et al., 2005; Topping et al., 2004), whilst regions with higher $NO_x$ concentrations favour nitrate formation (Hu et al., 2015; Parworth et al., 2015; Poulain et al., 2020; Schlag et al., 2016). The urban vs rural difference in relative mass composition is evident when comparing this study with the aerosol mass fractions observed in an urban site in Sydney (Keywood et al., 2016) in which high nitrate fractions were observed during most of the campaign.

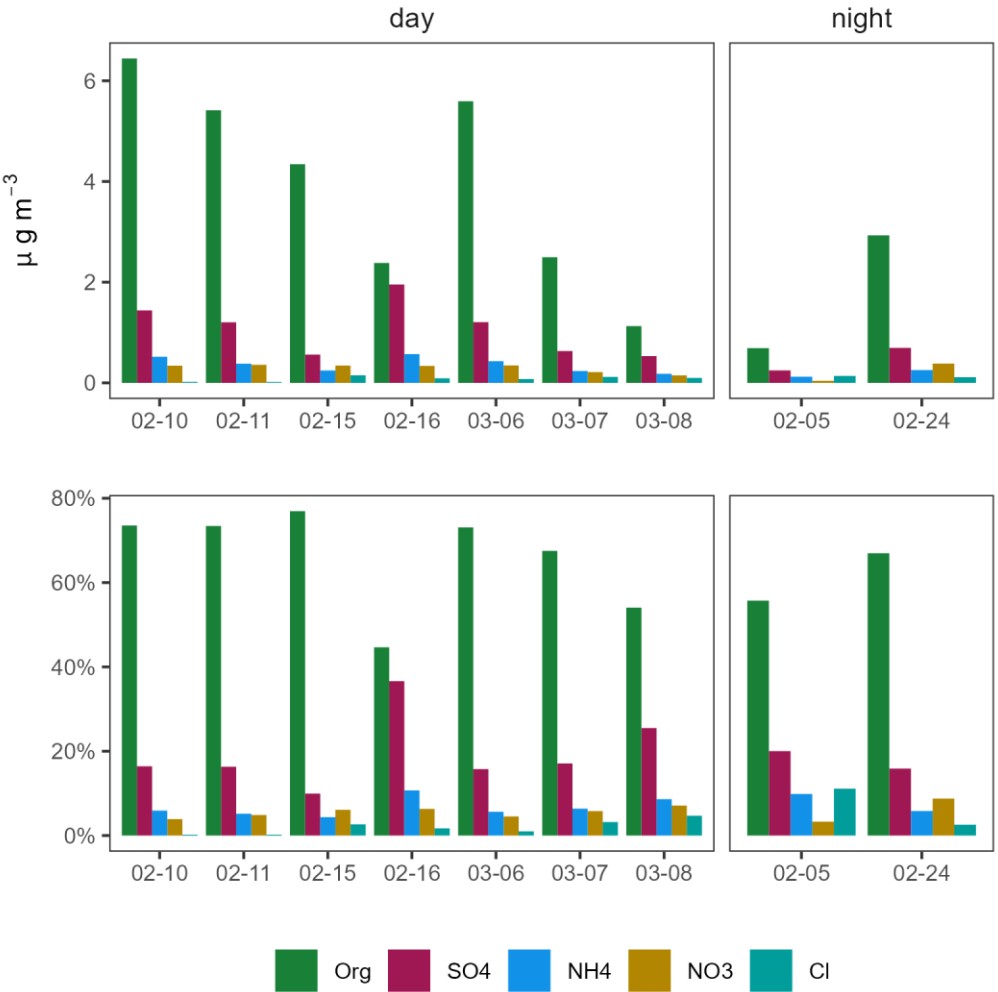

*Figure 9 Average mass for each chemical group and event on the top panels. The bottom panel presents the percentage contribution to the mass of each of those fractions based on the average value presented above. Org = Organics; SO₄ = sulphates; NH₄ = ammonium; NO₃ = nitrates and Cl = chlorides.*

## 4. Summary and Conclusions

Here we present aerosol concentration and composition data, VOCs and air pollutant concentrations collected during part of the COALA-2020 campaign including data from 5th Feb to 17th March at a rural site south of Sydney, Australia. This period followed the Black Summer fires after heavy rainfall cleared the smoke, offering insights into atmospheric processes under clean background conditions.

The atmosphere during the sampling period was classified as highly reactive with particle formation identified on more than 39% of the sampling days. Like previous studies, daytime NPF events coincided with the arrival of anthropogenic plumes at the site, suggesting their role in initiating particle formation. The positive relationship between monoterpene concentrations and both $PM_1$ organic aerosol mass and $CN_3$ suggests a direct relationship between biogenic emissions and organic aerosol formation.

The change between gas to aerosol phase was indirectly analysed through the evaluation of the conditions leading to NPF events. This analysis showed how $SO_2$ plumes impacting the site drove NPF. The particle growth rate was dependent on available VOCs in the atmosphere and OH availability, also enhanced during periods with higher relative humidity and multiple intrusions of $SO_2$ and $NO_x$ plumes producing particles larger than 100 $nm$.

Night-time events were attributed mainly to oxidation with ozone. Although most of the night-time events showed the influence of monoterpene ozonolysis on NPF events, our data was limited and we acknowledge that other factors may have influenced nighttime NPF.

The COALA-2020 campaign highlights the significant role of biogenic emissions, particularly monoterpenes driving NPF and isoprene enhancing particle growth in Southeast Australia. These findings contribute to a better understanding of local atmospheric chemistry and its potential impact on regional air quality and climate. However, longer-term observations are necessary to capture the full picture of seasonal variations and non-fire related extreme events.

**Supplementary Materials:**

**Author Contributions:**

The experiment design was made by Clare Paton-Walsh (Murphy) and Melita Keywood.

The data collection was done by Jack Simmons, Travis Naylor, Paton-Walsh (Murphy), Asher Mouat, Melita Keywood, Ruhi Humpries, Malcolm Possell and Jhonathan Ramirez-Gamboa.

The data processing to convert mass spectra to concentration of VOCs was done by Asher Mouat under the direction and supervision of Jennifer Kaiser.

The data analysis was done by Jhonathan Ramirez-Gamboa

The paper was written by Jhonathan Ramirez-Gamboa and Clare Paton-Walsh (Murphy).

**Funding:**

COALA-2020 was supported by Australia's National Environmental Science Program through the Clean Air and Urban Landscapes hub. Jhonathan Ramirez-Gamboa was supported during his PhD studies by a commonwealth funded University Post-Graduate Award at the University of Wollongong.

**Data Availability Statement:**

Data is available at PANGEA via the following links:

- VOCs: https://doi.org/10.1594/PANGAEA.927277
- Aerosol size distributions: https://doi.org/10.1594/PANGAEA.928853
- Condensations nuclei > 3 nm in diameter: https://doi.org/10.1594/PANGAEA.925555
- Cloud condensation nuclei: https://doi.org/10.1594/PANGAEA.928925
- Green-house gases: https://doi.org/10.1594/PANGAEA.927313
- Air Quality data: https://doi.org/10.1594/PANGAEA.929001
- Meteorological data: https://doi.org/10.1594/PANGAEA.928929
- ACSM data: https://doi.org/10.1594/PANGAEA.973272

**Acknowledgments:**

We are grateful to all who contributed to the COALA-2020 campaign. Particular thanks are due to all the staff at Cataract Scout Camp and research students and staff: Ian Galbally, Kathryn Emmerson, Gunashanhar Gunaratnam, John Kirkwood, Warren White, David Griffiths, Alex Carter, Alan Griffiths, Hamish McDougall and Graham Kettlewell.

**Conflicts of Interest:**

The authors declare no conflicts of interest.

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
