# Peer review of "Measurement report: New Particle Formation Events Observed during the COALA-2020 Campaign"

_EGUsphere, 2024_

## Author Comment (AC1)

The authors thank the referees for their helpful comments and suggestions. The original comments are below in black, and our responses follow in green. In response to the referees concerns on outdated references and discussion, we have updated our analysis based on the suggested literature. The changes include a more specific discussion on the potential causes of the observed NPFs reflected on current knowledge of clustering. We also differentiate the NPF from the larger growth sizes and endeavour in making the distinction when explaining our analysis.

Referee #1

My biggest concern is that while the data itself are certainly interesting, the results are not properly discussed and put into context of the developments in the field over the last several years. Even if no measurements with a nitrate CIMS are available to measure the nucleating species directly, also the existing data allows for a more thorough discussion. The main message of the manuscript is that NPF events are correlated with SO2 and NOx, but the given explanation for why these gases should lead to NPF is not adequate and in one case even wrong as discussed below. Also in the biogenic case it is mentioned that VOC oxidation plays a role for growth, but this role is not really specified. Currently the analysis is mainly pointing out correlation, but not investigating causation adequately.

We thank the referee for the referee for the comments. We acknowledge that the explanation of the gas to particle process was lacking detail in the manuscript, we have improved both the introduction and analysis with the suggested literature and clarified the role of the two main BVOC (monoterpenes and isoprene) groups we observed. In general, monoterpenes are related to NPF, as you highlighted the HOMs produced by monoterpene reactions lead to ULVOC or ELVOC which trigger nucleation.

In the case of SO2, we know that it can be oxidized to sulfuric acid, which is a prominent driver of nucleation, however, you do not really mention sulfuric acid in your discussion or interpretation. While sulfuric acid production via OH produced by photolysis is plausible for the early-morning event in Fig 2, this is not as straight forward for the nighttime event in Fig 7. Both events have elevated SO2, but one has photochemistry and one not, so the question is where would the OH be coming from in the nighttime case? It could be OH production via ozonolysis of monoterpenes, but you do not discuss this.

In order to get one step closer to the nucleation process, I would recommend to estimate the sulfuric acid concentration via proxies as e.g. described in Dada et al 2020 (https://doi.org/10.5194/acp-20-11747-2020), for which you have measured all necessary quantities (SO2, global radiation, condensation sink, RH). With these proxy values, you could then estimate whether sulfuric acid alone or in combination with ammonia leads to roughly the nucleation rates that you find in the field and in how far this would be comparable to other sites with sulfuric acid driven NPF and laboratory measurements.

We have estimated and added the H2SO4 proxy based on Dada et al 2020 for the rural scenario to the plots and the analysis helping us to better understand conditions for clustering formation. Unfortunately, we didn't have enough capacity to estimate and compare nucleation rates for each event. Nonetheless we consider the addition of H2SO4 proxy valuable to the analysis.

The discussion of SO2 driven nucleation in the section between lines 264 and 271 is not correct. You suggest that SO2 will produce $SO_4^{2-}$ radicals, which then trigger NPF. The $SO_4^{2-}$ time trace you use for that statement was however derived from the ACSM which measures particle composition of large (you state >40 nm) particles and not gas phase radicals. So the chain of events is flipped in your statement compared to what would actually happen in case of SO2/H2SO4 driven NPF (Oxidation of SO2 to H2SO4, then nucleation and growth, then, once the particles are larger than the cutoff of the ACSM, the $SO_4^{2-}$ signal in the ACSM would increase). Figure 4 and your statement in Line 264-266 also

points to that direction: First you have SO2, then CN3 increases after 120 minutes due to conversion of SO2 to H2SO4 and nucleation and then another hour later the ACSM SO42- signal increases as the particles have grown to large sizes. So everything fits together and flipping the order does not make any sense.

The analysis presented in this plot aims to show precisely what you stated here "First you have SO2, then CN3 increases after 120 minutes due to conversion of SO2 to H2SO4 and nucleation and then another hour later the ACSM SO42- signal increases as the particles have grown to large sizes" using time series analysis as we didn't have direct measurements of H2SO4 or the smaller particles. We have rephrased the lagging description so it now reads: "the $SO_2$ time series is lagged 120 minutes with respect to the aerosol data" and the paragraph as follows:

" …Usually, the $SO_2$ correlation with aerosol $SO_4$ takes longer to be significant. This is a potential indication of the order in which the chemical reactions happen. First, we will see oxidation of the $SO_2$ to $H_2SO_4$, then nucleation, and finally growth in mass when there is condensation or coagulation near CCN sizes. Using time series analysis as shown here can provide more evidence when the chemical mechanisms are known but observations of other variables are not available."

The studies of Wang et al 2019 and 2020b that you refer to in the section starting in line 291 to my understanding do refer to oxidation of SO2 to sulphate ions inside the aerosol phase in the context of haze formation and are thus not directly applicable to NPF that you study here.

We have removed the citations as we agree that the studies were not applicable to the conditions of our study.

In the case of NOx, you again give no clear explanation how, on a chemical level NO or NO2 are expected to trigger nucleation. While in the SO2 to sulfuric acid case there is a clearly establishes path for nucleation, for NOx it is much less straight forward. In the biogenic case NOx typically reduces nucleation rates compared to the no-NOx case. One reason for this would be that nitrate functional groups typically lead to higher saturation vapor pressures than e.g hydroperoxide groups.

If NOx emissions together with SO2 emissions do indeed contribute to NPF, then a potential mechanism could be the SA-HNO3-NH3 growth enhancement described by Wang et al 2020 (https://doi.org/10.1038/s41586-020-2270-4). The nitric acid could form via NO2 oxidation by OH.

After revising the literature, we agree that NOx does not enhance nucleation under the conditions of our study. We know H2SO4 has a direct role, and it alone could explain nucleation when there are pollution events. We wrongly attributed both NOx and SO2 a role in NPF. Attributing causation to a the correlation they have as we expect they come from the same source. We removed attribution of NPF to NOX but it is worth mentioning that it contributes to the larger sizes and SOA formation as seen in the ACSM data. The night-time event show below subsequently in the manuscript highlights nucleation and then growth where isoprene oxidation through nitrate radical oxidation pathway might be happening. We have adjusted the text accordingly.

[Figure]

2020-02-05 18:10:00  N

This could potentially be relevant to the event shown in Figure S7 at around 10 am. There is an initial rise in particle concentration both in CN3 and the SMPS and then at around 10:15 suddenly large particles appear. The important question is, is this an airmass change or really such a large growth inside the same air mass, which would then very nicely resemble the rapid growth event described in Wang et al 2020. I would recommend to investigate this further and check whether this potential rapid growth could be mostly inorganic, organic or both. (This is a prime example of very interesting data, but currently too little discussion)

We have added other panels to the plots to further support our analysis, among those wind direction and wind speed. For this particular event we know there is no airmass change so we are seeing real

rapid growth and tried to explained as stated here: *" The event on March 10 (Figure S6) shows high monoterpene concentrations that declined quickly just prior to the event being observed in the aerosol data. The aerosol growth phase is then observed to correlate with peaks in $SO_2$ and $NO_X$, as well as elevated levels of isoprene. Together, this suggests monoterpene ozonolysis initiated nucleation, followed by condensational growth via isoprene oxidation products."*

The authors do not report ammonia levels in this study, however the particles are containing ammonium as measured by the ACSM and another study to which some of the authors of the current study did also contribute (Phillips et al. 2019) showed high ammonia levels >5ppb for a site not too far away from the measurement site in this study. Wang et al report their mechanism for colder temperatures than are reported in the current manuscript. Maybe with higher concentrations than in Wang et al. you could overcome the effect of the higher temperatures. I am not saying that this is necessarily what is going on, just that more discussion of what is going on is needed.

The study mentioned by the referee was made under urban conditions and most of the ammonia was attributed to catalytic converter emissions. We deployed an open-path FTIR spectrometer over an extended path-length to obtain these measurements and such a technique was not possible within a forested area. Nevertheless, we would assume that NH3 concentrations would be significantly lower in this more rural area away from agricultural sources and with a significantly lower traffic load than Western Sydney (where the Phillips et al study was located).

Another important question is, where are the SO2 and NOx emissions actually coming from. You discuss this briefly and mention the Appin road and traffic emissions. Can you add wind direction and speed to the time series plots and indicate (eg with colors) when an airmass was influenced by road traffic. This would make the plots easier to interpret. Especially to distinguish between changes in chemistry and real growth on the one hand and simple air mass changes on the other hand. (In Figure S7 it looks like the sudden growth event at around 10:10 is within an airmass, but the sudden appearance of particles at around 22:40 might be caused by airmass change. Also at the vertical black line at 13:40, indicating the end of the event, it simply looks like the airmass has changed to a more isoprene dominated and less polluted air mass). Related to this, you could also look whether you have any known ammonia sources along the way, if you have high SO2/NOx

As mentioned above SO2/NOx is attributed to the same source impacting the site, likely mobile source emissions or to more distant coal-fired power stations. Either way ammonia would also be co-emitted with these compounds. Most of the pollution events are related to wind directions where the closest main road is but there are some other events (like the one mentioned by the referee) that might be coming from urban sites further away. There is a chance that there is some ammonia being emitted along with SO2 and NOx given the lax emission control in NSW. Referring to this event, there is airmass change but later to the time suggested by the referee and in fact wind speeds are relatively low. Nonetheless, the air mass might have changed at 12:00. Because we didn't have ammonia measurements during the campaign we don't mention it in the analysis but the text have been adjusted mentioning the potential influence oof distant coal fired power stations

2020-03-10 08:10:00 D

Another comment is that the manuscript tends to use imprecise or overly generalizing language. One example is from the introduction: "Secondary aerosols are produced via gas-to-particle transition, where reactive compounds in the atmosphere are oxidised to become low volatility organic compounds (LVOC). These compounds, along with sulfuric acid vapour are often involved in the nucleation process promoting clustering (e.g., Yu and Luo, 2009)."

As it is currently formulated it would mean that LVOCs are involved in the nucleation process, i.e. initial cluster formation. This is however not the case, as ELVOCs or ULVOCs are typically needed for this, which were described only after 2009. In this context I would recommend to more clearly separate the process of SOA formation, which in traditional understanding needs preexisting seed particles, and New-Particle-Formation (which means initial cluster nucleation and early growth), as both processes have very different requirements for the involved molecules in terms of saturation vapor pressure. Since the title of the manuscript contains New-Particle Formation, I would recommend to focus

especially on this vs SOA formation (which you do not mention in the later part of the manuscript anyway).

The field of biogenic driven New Particle formation is rapidly evolving, however many of the references that are cited in the introduction regarding this topic are from the 2000s, which was before the discovery of ELVOCs/HOMs and pure biogenic nucleation. I would recommend to use newer literature to make your point in a more precise way, eg Kirkby et al 2023 (https://doi.org/10.1038/s41561-023-01305-0) or Zhao et al 2024 (https://doi.org/10.1038/s41586-024-07547-1), Bianchi et al 2019 (https://doi.org/10.1021/acs.chemrev.8b00395), or others.

Another example of imprecise or out-of-date wording is the section from line 62-70: The first sentence implies in a sense that multi-generation oxidation is key to low saturation vapor pressure and refers to Kiendler-Scharr et al. 2009, which was before the discovery of autoxidation for atmospheric terpenes, as well as OH recycling pathways (eg Taraborelli et al. 2012). HOM formation from monoterpenes and the related autoxidation is a 1$^{st}$ generation process with only one ozone or OH attack and one RO2 termination reaction. Only isoprene with its two double bonds is capable of 2$^{nd}$ generation chemistry and this is an open question how autoxidation and 2$^{nd}$ generation chemistry balances in this case, but you only very broadly speak of VOC oxidation.

The main reason why isoprene scavenges nucleation is not competition for OH, as stated by Kiendler-Scharr 2009, but interference in monoterpene-RO2 termination reactions with isoprene RO2 radicals that suppress C20 production in favor of C15 production, which are weaker nucleators. Since isoprene chemistry is OH dominated and monotperene HOM formation ozone dominated, higher OH leads to a higher isoprene suppression effect.

The last sentence in this paragraph about high levels of SO2 and VOCs is somewhat isolated and unspecific.

In general, many of the discoveries related to HOM/ELVOC formation and their implications for NPF are missing from the introduction and discussion, even if this directly affects the main topic of the manuscript.

All the previous comments refer to out of date or imprecise wording/explanation when explaining the VOC oxidation pathways and their roles in NPF. To address it we have updated the manuscript introduction and discussion with up-to-date literature as follows:

[revised manuscript text omitted]

Overall I would suggest to improve the figures with more traces that are relevant to NPF (temperature, H2SO4 and HOM proxies, wind direction, etc) and based on these figures arrive at a more thorough interpretation and chemical classification of the events.

One thing that could also help is to sum up the smps channels to get a total particle count above 14 nm. You can then subtract CN3-CN14 and derive a time trace of nucleation mode only particles. This should give you a nice indication of when NPF is happening exactly, as the definition purely based on smps data tells you only when particles where growing above 14 nm, which depending on growth rates could take several hours. (I am not suggesting to get rid of the current estimation, but to add the CN3-CN14 method to it)

The ordering of panels in the time series plots is unintuitive, as you go from gas phase precursors to chemical composition of large (>40 nm diameter) particles back to small particles (CN3). I would suggest to put the ACSM panels at the bottom.

The time series plots have a large amount of white space between the panels, which makes the space available for actual data very small. I would suggest to place all panels directly on top of each other and only have x-axis labels on the lowermost panel to save space. I would also suggest to include two panels (one for Temp and RH and one for wind speed and direction). You might also include global radiation, ozone and sulfuric acid and HOM proxies, in order to make the story for the NPF event as comprehensible as possible.

We have added H2SO4, winds, CS, ozone and MTp*o3(when available) to the plots. We also reorder the plots following the advice to reflect the evolution of the particles. We have also added the CN3-CN14 data stream to the analysis, as suggested by the reviewer.

How is it possible to sustain 1-2 ppb of NO during the night (e.g in Fig 2 or S7), when no NO2 to NO photolysis is taking place, traffic emissions might be lower than during the day and 5 ppb of ozone should oxidize NO to NO2 within 7 minutes? Are these values background corrected?

**No, the NO values are not background corrected. This text has been added to the description of these instruments in the methods section:**

**"All NSW air quality monitoring stations are accredited by the National Association of Testing Authorities (Australia), however it should be noted that these instruments are targeted at regulatory standards and are not research-grade. In particular, this means that measurements made close to the detection limits are likely to be inaccurate and should be interpreted as indicative measures rather than accurate quantitative measures of atmospheric concentrations."**

We also added this to the caption of Figure 2:

"Note that the NO values are close to detection limit and look biased high and hence should be interpreted as an indicative rather than accurate quantitative measure of atmospheric concentration. "

**Specific comments:**

line 60: can you mention the key condensing species?

The introduction was mostly rewritten but we were referring to ULVOC, ELVOC and their role in the NPF process. This has been added and the difference with LVOC and their role in SOA production has been clarified.

line 182: You state that you measure at a rural area with relatively low anthropogenic influence, but a large number of the NPF events are influenced by NOx and SO2, so I would not call it low anthropogenic influence.

We agree that this might be not considered pristine, so we have reworded it as a site with some anthropogenic influence

line 197: The term "density of particles" is misleading. Density refers to the mass to volume ratio of the solid or liquid particles in units of kg/m3, you seem to refer to the number concentration of particles, as you point to the CN3 measurements.

Reworded to particle number

In Figure S2 you show an interesting event starting at 11 am. Here the ozone trace would be very helpful, as it looks like you have strong biogenically driven or enhanced NPF in addition to H2SO4 driven NPF. Can you provide a proxy for HOMs like monoterpene*ozone timetrace as well as the sulfuric acid proxy. More than 1 ppb of monoterpenes should lead to strong nucleation with typical levels of ozone. Can you derive a nucleation rate for this event and compare them with laboratory values, eg Kirkby et al 2016? (At least to get a sense for the order of magnitude)

We have added the proxies as suggested. These are the higher values of the HOM proxy observed in the analysis. Although there are new particle bursts observed they might have been quickly coagulated on the preexisting particles. This correlates well with the higher condensation sink, higher particle number in the larger bins of the smps data and the increase in aerosol mass.

Figure 8: here again the ozone trace and the HOM proxy monoterpene*ozone would be interesting, as it looks like as isoprene is decreasing monoterpene driven NPF is appearing. This would be actually confirm the isoprene suppression effect on NPF. Additionally to ozone, in this case it looks like NO3

driven oxidation of monoterpenes might be important, as NO2 is very high, so NO3 will certainly also be high after sunset. You can compare this to Yan et al 2020 (https://doi.org/10.1021/acs.est.3c07958), which report a suppression effect of NO3 on NPF.

We agree with the potential of isoprene suppression effect. We rephrased the comments on this event as: "The event on Feb 5$^{th}$ might be showing a combination of different factors at play. First, the monoterpenes and ozone levels can be triggering nucleation as observed during the daytime events, but there is a slight increase in monoterpenes concentrations potentially driven by slower wind speeds and less mixing volume at night. Second there is the isoprene continuous decrease. There are no enhancements of MACR+MVK so we could speculate that isoprene is being oxidised by the $NO_{3-}$ pathway. This is encouraged by the slight increase in the Nitrates fraction observed with the ACSM around midnight. Later that day around 4:00 am there is a second burst of small particles that follow the same pattern of monoterpenes/ozone."

Figure 9: Do you know why the CN3 levels are relatively low during the NPF event, lower than before, whereas the SMPS clearly shows an increase in particle number. It looks like the actual nucleation starts already well before the green line and the growth rate is rather slow, so the small particles might have coagulated away already. You could check this with the CN3-CN14 method outlined above.

There was an air mass change at the time that could be explain the difference observed in both CN3 and SMPS data. Note that this plot was moved to the supplement material while preparing the new manuscript version.

line 288: Unclear if you mean that isoprene or SO2 are promoting NPF. You further state that "This event shows how even when VOCs available if there is no $SO2$ in the atmosphere (13:00) the particle formation will substantially decrease". Not all VOCs have the same NPF potential, as pointed out above, this needs a more in-depths discussion.

We agree, we reworded all the analysis to differentiate between isoprene and monoterpenes influence on both NPF and growth to SOA. This line was reworded as follows:

*"The event on March 10 (Figure S6) shows high monoterpene concentrations that declined quickly just prior to the event being observed in the aerosol data. The aerosol growth phase is then observed to correlate with peaks in SO$_2$ and NO$_X$, as well as elevated levels of isoprene. Together, this suggests monoterpene ozonolysis initiated nucleation, followed by condensational growth via isoprene oxidation products.*

*"*

line 387f: I would not expect NPF to significantly increase PM1 mass, as the mass is dominated by the larger particles. You can still have significant NPF with a strong increase in ultrafine number concentration without affecting the overall mass concentration much.

We agree. There was a redaction mistake, this PM1 mass is referring to the PM1 mass as the ACSM aerosol mass measured during the campaign not total PM1 mass. The ACSM data showed direct changes after NPF and growth events in many of our observations.

**Technical comments:**

Figure S4: You mention MACR+MVK in the figure caption, but you do not show the data in the plot. You also mention how VOCs are consumed at around 22:00 in line 339, but you do not show VOC data.

There was a mistake in the caption and VOC data was not available for this event

Figure S6: Can you spell out PAR (PPFD) in the caption?

Added to the caption

In summary, the manuscript describes a really nice dataset, but more analysis regarding causations and chemical pathways for NPF is needed.

Referee #2

This manuscript describes field observations of gas vapours, particle number distribution and composition in an Australian site at Cataract Scout camp. These measurements are utilised to find correlations of SO2, organic vapours with aerosol formation events. While such measurements are indeed valuable, since new particle formation measurements are rare in the southern hemisphere, and therefore potentially warrant publication.

However, the manuscript itself is very poorly prepared. It is clear that the authors did not follow the recent proceedings of new particle formation, isoprene oxidation and relevant topics. The discussions, references provided in this manuscript are rather out-of-date, and lots of them are even wrong. There appears many grammar and structural issues with the writing of this manuscript which make it extremely difficult to grasp useful information from this manuscript. The connections of the paragraphs are rather poor - it often occurs that the discussions jump from one point to another without any explanation. The same also applies internally to the same paragraph.

Scientifically, there are many flaws in the discussions of this manuscript.

1)The authors frequently mix up basic concepts, such as suggesting: 1) aerosols only have three modes and VOC oxidation by nitrate anion (NO3-) instead of nitrate radical (NO3), etc.

We thank the referee for the comment. Although aerosols are usually classified in 4 modes, coarse mode is not relevant to the analysis and hence we didn't include it in our introduction or analysis. Corrected the error naming NO3- instead of NO3

From their measurements, the authors suggest that SO2/NOx have strong impacts on their observed NPF events. While the impact of SO2 is understandable from the literatures, since it leads to the formation of H2SO4, which subsequently leads to particle formation, the role of NOx is mere speculation. The authors propose that under low NO conditions that organic oxidation might be enhanced (Nie, 2023) but their measured NOx concentrations are rather high compared to clear environments. Therefore, it is less likely the case. I should also note here that the H2SO4-HNO3-NH3 mechanism proposed by the referee #1 might not be relevant here since this mechanism is primarily important for cold environments, and there is no evidence that this will be important for boundary layer conditions.

We agree. This suggestion was given based a correlation instead of causation. As we think SO2 and NOx have the same source. We removed attribution of NPF to NOx but highlight that it has a role in larger particle growth in the Aitken and accumulation modes particularly at night through isoprene oxidation by nitrate radicals (Rongrong Wu, 2021) https://doi.org/10.5194/acp-21-10799-2021.

3) The authors propose in this manuscript that isoprene is important for particle formation at their site but clear evidence is lacking. There were sufficient literatures suggesting that isoprene might inhibit aerosols formation under boundary layer conditions and the mere correlation provided in this study does not in any way challenge that. I suggest the authors to read these recent literatures and base their discussion on these studies instead of proposing correlation as the cause-and-effect. It should also be mentioned that the authors consistently measure high levels of monoterpenes in their measurements, which by themselves might be sufficient for triggering aerosol formation from monoterpene and H2SO4 system. I recommend the authors to read, for instance, Lehtipalo et al 2018 (10.1126/sciadv.aau5363).

As stated in the answer the comment of referee 1 we agree with this. We have updated both the discussion and the introduction with up-to-date literature both regarding NPF and growth of particles in Aitken and Accumulation mode.

4) the nighttime event presented in this study is very interesting. However, it should be noted that during these events no clear gas transitions, e.g., VOC, SO2, NOx, are consistently observable in all cases. In fact, in cases presented by Figure 8 and 9, these vapours did not change at all. This is rather surprising since it either suggest 1) the particle formation is not a particle formation but the meteorological conditions change at that time or 2) some other vapours are contributing to the particle formation events. To answer this question, the authors have to do more through investigations which include meteorological data, such as wind directions, back trajectory etc. to help them to interpret their observations.

As stated in the answer to referee #1 we have included meteorological data and other proxies to improve our analysis. In this case, ozone was helpful to understand what processes might be involved in the NPF event. In regard to the night time event the text in the manuscript reads as follows: " *The event on Feb 5th might be showing a combination of different factors at play. First, the monoterpenes and ozone levels can be triggering nucleation as observed during the daytime events, but there is a slight increase in monoterpene concentrations potentially driven by slower wind speeds and less mixing volume at night. Second, isoprene is observed to be decreasing throughout the event. There are no enhancements of MACR+MVK so we could speculate that isoprene is being oxidised by the nitrate radical pathway. This is encouraged by the slight increase in the nitrate fraction observed with the ACSM around midnight. Later that day around 4:00 am there is a second burst of small particles that follow the same pattern of monoterpenes/ozone. The monoterpene ozonolysis is also seeing in other night events (see Fig S8-S10)."*

Finally, the quality of the present manuscript prevents me from giving detailed comments which I hope the authors can address in their next version. While I have concerns about whether the authors can sufficiently address my suggestions and improve the quality of their manuscript to an ACP standard, I would recommend giving them a chance to improve their manuscript because of their valuable and rare measurements. It should also be noted that this manuscript should be categorized as a "measurement report" instead of an ACP article.

Major comments:

There is no clear storyline in the introduction. Many paragraphs are discussing their own independent evidence with very little connections between them. Even worse, within each paragraph, it reads like pileup of unconnected evidence and occasionally rather out-of-date, even wrong statements. This is very clearly seen from e.g., the lines 44 to 70, which I will give example as below:

L44-52: This paragraph mainly discusses the secondary aerosol formation processes, CCN and common knowledge of size distribution definitions. This paragraph contains multiple essential and basic errors which led me to believe that either the authors did not take their time to read the recent literatures. First of all, Yu and Luo didn't mechanistically find nucleation of sulfuric and/or organic vapours; therefore the references to their paper to show the involvement in NPF is inappropriate. Better references are needed here. Second, clusters and ultrafine particles are not the same thing, so lease remove "(ultrafine particles)" as surrogate of clusters. Additionally, aerosols are commonly discussed in four modes and not three (coarse mode, >1000 nm) which is a common knowledge in the field. I should also mention that I do not understand why one has to define the aerosol modes in this paper? Since the paper itself has nothing to do with classifying size distributions and also because the potential readers of this manuscript would surely know the four modes.

L53-60: It is rather disturbing to read through this paragraph. EVERY SINGLE SENTENCE in this paragraph is hardly connected to one another. I give a summary of each sentence in their respective order as below: 1) BVOC is important, 2) Monoterpene has higher SOA yield than isoprene, but there is more isoprene, 3) isoprene - NO3 has higher SOA yield than isoprene - OH, 4) isoprene SOA yield in low NOx conditions is higher than expected. First, there lacks connection with the previous paragraph, and it feels that this paragraph comes from nowhere. In the second sentence, the monoterpene and isoprene have not been properly introduced yet and it directly arrives at the point that their yields and emissions are compared. In the third sentence, main channels of chemical reactions of isoprene have not been explained yet and the product yield of NO3 is compared with OH. Additionally, the sentence does not make any connection with the previous sentences and the information provided is rather outdated. Isoprene oxidation primarily occurs during the daytime through OH oxidation channel, why the higher yield at night from NO3 than OH matters at all in this discussion? In the fourth sentence, why it suddenly jumps into discussing isoprene SOA yield at low NOx conditions, especially considering the previous sentences discusses higher SOA yield from isoprene - NO3 channel? Last but not the least, what this paragraph eventually wants to tell the reader? What is the conclusion of this paragraph and the association with this manuscript?

Lines 61-70: Again, no connection with the previous paragraph. The statement "To form key condensing species, multiple oxidation steps must happen to the original VOC molecule" is quite out-of-date and ignores the auto-oxidation channels that have been discovered in the gas phase since more than 10 years ago. The final sentence is detached from the rest of discussion.

As stated in the answers to Referee #1, we have updated and improve the way the introduction outlines the processes of aerosol formation and what the role is of different chemical compounds, closing with highlighting of the problem of understanding both aerosols and biogenic emissions in the Australian context. Please refer to the answer to Referee #1 to read the updated version of the introduction.

Minor comments:

Figure 1: are the authors sure that they are free to share Google Earth's pictures as they are here?

We have changed the map for an opensource option.

L63: define OVOC

Defined in the text

L181-183: how the frequency alone tells you about the level of low anthropogenic influence? In the least case, some back trajectory analysis would be required to even remotely conclude on this.

The frequency in this paragraph is talking of the number of NPF events per days of observation. The statement of having low anthropogenic influence came from the location and the distance of the main urban sites. Nonetheless we agree that it might not be as pristine as originally thought, so we have reworded it as a sampling site with some anthropogenic influence.

L192-194: it is not clear what this sentence is describing, please rephrase.

Rephrased as: " . As an example, the event on Feb 11$^{th}$ 2020, presented in **Error! Reference source not found.** shows how after there was a first $SO_2$ peak coming to the site before 8:00 am there is a small burst of nucleation. The $H_2SO_4$ concentration is enough to trigger bursts of nucleation prior to the times selected by the NPF identification method, as shown in the $CN_3$-$CN_{14}$ panel. This event did not show a quick early growth like several other events in the record possibly due to the early morning start time when photochemical activity is reduced and there are not many VOCs to be oxidised and thus, accelerate the growth process."

L194-195: refer to these "several other events" in the SI.

We thank the referee for the comment and agree to some extent that looking at multiple plots in the SI is not ideal but having all plots from the events in between the text will cut the flow of the narrative.

L196-198: rewrite the sentence.

This is a continuation of the comment on L192-194. It was mostly rephrased as: "The highlighted area is showing a growth stage around one hour after the nucleation explained by the changes in atmosphere composition. Once the temperature starts to increase, enhancing the VOCs emitted, and there is more OH available in the atmosphere, there is a higher number of particles in both nucleation and Aitken mode. This difference is reflected in the peak captured in the $CN_3$, the increasing difference of $CN_3 - CN_{14}$ and condensation sink data. Notice how the condensation sink continuous to increase along with the aerosol mass and particle size. Sometime before 14:00, the wind changes direction almost 180° so it is likely we are seeing growth of the same air mass that initially crossed over the site a few hours earlier to the larger sizes."

L206: were --> where

The plot was modified and no longer applies

L208: it is clearly missing something. I do not understand how one jumps from L199 to L208 without further evidence presented. The authors should show their results which lead to " SO 2 appears to only affect daytime events, while NO x seems to have a shared role in both daytime and nighttime events" first.

This phrase was removed during the preparation of the last version of the manuscript.

L219: what plumes the authors refer to?

The plumes refer to air masses coming from pollution sources. This has been rephrased in the manuscript as "pollution plumes" for some clarity

L233: nitrate radicals (NO₃) not nitrate anions.

Corrected

L233-236: rewrite the sentence.

Rephrased as: "*These reaction pathways might include isoprene oxidation by nitrate radical ($NO_3$) oxidation path during the night (Wu et al., 2021b), monoterpene ozonolysis and condensation over previously formed clusters(Liu et al., 2023; Wang et al., 2023), or oxygenated VOCs (OVOCs) brought to the site and condensed on formed seeds or starting nucleation(Bianchi et al., 2019; Higgins et al., 2022). Some of these processes were observed during the campaign and will be further explored on the nighttime events section..*"

L243-245: rewrite the sentence. Also note that the bracket ")" is missing.

1.  Rephrased as: " *The hours with high VOCs concentrations and higher oxidation capacity in the atmosphere ($OH$ concentrations are assumed to be higher during the hours with higher PAR) have higher particle number concentrations and generally guaranteed growth up to the accumulation mode..*"

L268: SO4(2-) is not a radical

Rephrased the paragraph refer to the next comment.

L269-271: It is not clear to me what the authors are talking about. SO2 forms H2SO4 first, then nucleation and eventually forms SO4(2-). How do they derive their order of chemical processes otherwise?

As stated in response to Referee #1, this analysis aims to provide more evidence on what is happening from a pure data analysis perspective. The paragraph reads as follows: "

To interpret Figure 4we can use the event on February 11ᵗʰ (black line) as an example. Here the correlation between $SO_2$ and $CN_3$ becomes significant (at $|r| > 0.5$) if the SO₂ time series is lagged 120 minutes with respect to the aerosol data; and the correlation between $SO_2$ and $SO_4^{2-}$ becomes significant after 3 hours. This means that if we move the $SO_2$ time series two hours forward it will be better correlated with the particle number concentration, accounting for the reaction time of $SO_2$ to produce $H_2SO_4$ and enhance/trigger the particle formation under the conditions in the atmosphere at the time. Usually, the $SO_2$ correlation with $SO_4^{2-}$ needs a longer lag time to be significant. This is a potential indication of the order in which the chemical reactions happen. First, we will see oxidation of the $SO_2$ to $H_2SO_4$, then nucleation, and finally growth in mass when there is condensation or coagulation near CCN sizes. Using time series analysis as shown here can provide more evidence when the chemical mechanisms are known but observations of other variables are not available.."

L291-302: the authors clearly ignored the recent proceedings of direct condensation of oxidised organic molecules and tried to refer to a nuance channel to explain why their measured NO2 should be related to NPF.

As stated before this was removed from the analysis.

L349-350: the authors clearly do not know what they are talking about. They first propose that monoterpene oxidation products may contribute to the slow growth and then explains it that MACR (from isoprene) needs to be oxidized to produce aerosols.

Refer to the next comment

L352: the authors should check what the low NO concentrations Nie(2023) refers to. Does that align with the constant ppb levels of NOx in Figure 7?

This part of the manuscript was rephrased considering the added proxies, tracers and up to date literature as follows: *"When there is no $SO_2$ in the atmosphere but there are high enough VOC concentrations, there can be particle growth when dimers $C_{15}$, produced by further OH oxidation of isoprene products, condensing onto smaller particles. Growth was observed during the February $10^{th}$ event (**Error! Reference source not found.**) and may be related to the condensation of these dimers. $CN_3$-$CN_{14}$ data suggest that there were small bursts of nucleation but where rapidly coagulated on already formed bigger particles. The night time nucleation observed at midnight is related to an airmass change and might be the result of a combined effect of monoterpene ozonolysis and subsequent OH production after Crieege intermediates decay (Lester and Klippenstein, 2018). Once the OH is available it can produce $H_2SO_4$ and enhance the nucleation process. In this case the increases in organic and sulphate mass shortly after the ozone depletion and the increase in CS indicate a growth in existing particles that is visible in the larger sizes in both particle numbers and mass."*

L363-364: what does "Like isoprene, the availability of monoterpene in the morning may determine how fast a NPF event can occur after SO 2reaches the site." mean? The previous paragraph was mentioning monoterpene oxidation etc, except the false reference to MACR. Can the authors be more specific with what they are describing?

Refer to the next comment.

L365-367: the authors should compare the OVOC yield of monoterpene from the OH and O3 channels and isoprene oxidation from OH channel before they derive such conclusions. These discussions are not quantitative, difficult to understand and potentially even qualitatively wrong. They have measured the needed vapour concentrations and should not stop at discussing qualitative phenomenon.

This whole paragraph was rephrased as follows: *"The availability of monoterpenes increases the likelihood of NPF. Autooxidation can start NPF and it also enhances nucleation when $SO_2$ is available in the atmosphere. Although monoterpenes are quickly oxidized by OH resulting in relatively short lifetimes compared to isoprene (Atkinson, 2000; Atkinson and Arey, 2003), the ozone levels observed during the campaign are enough to promote ozonolysis and nucleation when there is no OH competing. Australia experiences an isoprene dominated atmosphere (Emmerson et al., 2016; Ramirez-Gamboa et al., 2021) so the chemical balance in the atmosphere can rapidly change, particularly in the hotter seasons when more isoprene is emitted. Isoprene oxidises mainly through the OH pathway to more stable compounds; usually MACR and MVK are used as tracers to determine which path and under what conditions isoprene is oxidised. MACR is oxidised to heavier OVOCs that eventually condense and promote SOA formation in the larger sizes but these compounds also suppress NPF (Heinritzi et al., 2020; Link et al., 2021) as previously discussed in the event in **Error! Reference source not found.**. "*

L370-379: 1) how do the authors conclude: "In the absence of monoterpenes but presence isoprene the particle formation may be of smaller magnitude and the formation may occur at a slower rate."2)

how do the authors conclude: "MACR is oxidised to heavier OVOCs that eventually condense.". In more recent isoprene oxidation studies, this statement is clearly wrong.

*The first part of the statement is wrong and was corrected as show in the previous comment attributing the NPF to monoterpenes even at low concentrations levels as observed in the campaign. 2) Heinritzi et al., 2020 clearly states how monoterpenes indeed produce C20 and how isoprene products favours C15 after their respective oxidations. They also show how C15 dimers enhance growth above 3.2 nm, but can suppress growth below that by increasing isoprene derived RO2 that reduce the C20 that effectively reduce the HOMs reducing early growth.*

L398-399: the authors hypothesize that "Another factor possibly influencing the NPF events at night may include the early night VOC accumulation in the residual planetary boundary layer." without evidence. In fact, their measurements in Figure 8 clearly see decreasing trend of VOCs. Can the authors elaborate on this?

*We have removed this part from our analysis as we didn't have the conditions explored in other studies during the NPF events.*

L440-441 and L449-450: how the authors derive that isoprene is enhancing the aerosol number while Heinritzi (2020) suggested otherwise. The evidence in this study is rather a correlation at maximum and does not pinpoint the underlying mechanism. It should also be noted that the monoterpene concentrations measured in this study is itself enough for triggering particle formation events.

*We agree with the referee here. This statement has been removed and the citation applied in the correct context in which we suggest there was isoprene suppression of a monoterpene related NPF*

---

## Author Response (AR2)

We gratefully acknowledge the reviewers for their valuable feedback and suggestions, which have been instrumental in refining this paper. Their original comments are provided below in black, with our corresponding responses in green.

Referee #1

I appreciate the addition of the Monoterpenes*O3 trace in the figures, however, here a unit is missing on the axis labels.

Thanks for the comment we have added units [ppb*ppb] where pertinent

L 15+16: doubling of „play an important role" + what is meant by the second part of the sentence? Typically it is more the other way round that gas phase concentrations affect the aerosol levels. But are you talking about chemistry in/on the particle?

Fixed the typo. Now it reads: "…and by affecting trace gases through chemical reactions occurring in and on aerosol particles."

Line 368. The wording of point 3 is not clear and a bit confusing

This discussion was rephrased as:

"During the COALA-2020 campaign, many events, such as the one on February 16$^{th}$ (Figure S3), exhibited elevated gas-phase SO2. The availability of monoterpene to form highly condensable ULVOC/ELVOC is crucial in the observed events. While the oxidation products of isoprene can also condense on pre-existing particles (Stangl et al., 2019), the dominant pathways and their efficiency are likely driven by monoterpenes. Although VOC data was not available for February 16$^{th}$, the consistent diurnal profile of VOCs observed throughout the remaining dataset (Figure S5) suggests enhanced monoterpene and isoprene availability during the daytime. Under these conditions of available BVOCs, particle growth was frequently observed, suggesting a contribution from condensed organic material. As the night approaches and BVOC emissions decrease with temperature, the remaining OVOCs can undergo further oxidation, forming less volatile species that are more prone to condensation on existing particles. However, the limited availability of VOCs after their consumption (estimated around 22:00 based on diurnal cycles in Figure S5) likely limits further growth."

Line 606: "Although monoterpenes are quickly oxidized by OH resulting in relatively short lifetimes compared to isoprene (Atkinson, 2000; Atkinson and Arey, 2003)". The reaction rate constant (IUPAC preferred values) of alpha pinene + OH is roughly half that of isoprene + OH for, so at any given OH level, the isoprene lifetime would be half the alpha pinene lifetime with respect to OH. This sentence is however giving the impressions that monoterpenes react faster with OH than isoprene. Can you rephrase it or be more precise here?

This discussion has been significantly revised and now reads as follows:

[revised manuscript text omitted]

The authors discuss isoprene and monoterpene contribution to particle growth without chemical ionization measurements. They claim isoprene contributes more which needs evidence. This paper fails to link previous quantitative information with their current paper and more work needs to be done on discussions throughout this paper.

We appreciate the reviewer's concern about the lack of chemical ionization measurements and the previous emphasis on isoprene. We have addressed this by rewording the discussion to focus more directly on our observations and have reduced the asserted contribution of isoprene NPF role based on the provided references.

For the section 3.5, the night-time event is interesting, as I already mentioned. But the discussions provided are currently not convincing enough. If monoterpene is the cause of the NPF (including proving OH and sCI for SO2), then why the mid-night in Figure 6 did not observe the same NPF event since the monoterpenes are even higher there? Additionally, since the CS has increased substantially, I'm surprised the ACSM did not see significant increase of different elements. The sulfate decreased and the organic is decreased. There are only minor changes in the ammonium, chloride and a bit later nitrate. So first of all, why ACSM measurements did not observe a significant change and 2) why these elements increase instead of sulfate and organics? This might be leaving to a more robust discussion. Also, the authors should consider the possibility of a transported event instead of a local NPF for this night time event as evidence of local NPF is rather lacking.

We appreciate the reviewer's insightful comments on the night-time NPF event. Upon further investigation, we have identified one event that was likely due to a shift in wind direction and have removed it from our analysis. The remaining discussion of night-time events has been modified to be more observation-based, and we highlight the potential involvement of NO2 chemistry. Regarding the ACSM data, the lack of fully concurrent measurements across all events limits our ability to provide a comprehensive explanation for the observed trends. However, we find it interesting that the nighttime NPF frequency in our study are significantly lower compared to previous studies in the vicinity, which could be related to nitrate suppression observed during our campaign.

The night time section now reads as:

"We observed three nighttime events during COALA. Unfortunately, none of these events coincided with all data sets being collected which limits our ability to discuss the reactions driving the nighttime events. Consistent between all nighttime events is an increase in particles ($CN_3$), elevated $NO_2$, and an increasing condensation sink. Unfortunately, the $NO_x$ instrument available in this study was not ideal for this type of measurement for several reasons: it is not designed to be sensitive to the low $NO_x$ levels observed in rural areas; it is not capable of separating $NO_x$ from $NO_y$; and it was set up to calibrate in the night hours between 1:00 and 2:00 every day. Nonetheless, during the night-time events the particle size distribution data and the $CN_3$ data showed particle formation and growth from nucleation to Aitken modes when there were considerable increases of $NO_2$ and simultaneous decreases in ozone.

When VOC data are available, monoterpene concentrations were moderate and increased steadily during the event ($5^{th}$ Feb and $9^{th}$ March). Isoprene was high at the start of the event on $5^{th}$ Feb, (see Figure 8) however the sudden decrease in isoprene concentration likely coincides with sunset on that day. When aerosol composition data was available ($10^{th}$ Feb) aerosol organic, nitrate and sulphate concentrations increase during the event. When ozone data were available, concentrations decreased slightly during the course of the event.

The frequency of nocturnal events observed in this study is lower than observed previously at a nearby location (Tumbarumba a eucalypt forest site located 300 km to the SE of Cataract (Suni et al., 2009)), where in the summer of 2006, nocturnal NPF events were observed on 32% of the analysed nights and occurred 2.5 times more frequently than daytime events.  Simulating the NPF at Tumbarumba, Ortega et al (2012) was able to reproduce the observations from Tumbarumba by ozonolysis of 13 - carene to initiate nucleation and a-pinene to grow particle diameters. Ozonolysis of limonene was found to contribute to both nucleation and aerosol growth.  The lower frequency observed in our study may be linked to the apparent initiations of nucleation by $NO_2$, which nocturnally can react with $O_3$ to form nitrate radicals.   Li et al. (2024), suggest even trace amounts of $NO_3$ radicals suppress the NPF."

Minor:

L18: It is hard to argue that vapor condensation and growth are truly distinct processes. Perhaps it would be better to simply refer to them as "nucleation and growth."

Changed as suggested

L19-20: This is gas-phase oxidation and not heterogeneous processes...

Removed the heterogeneous process attribution

L27-28: OH is not just influenced by relative humidity and there are many other factors controlling its concentration. Please remove the brackets.
Removed the brackets

L31: why wildfires are mentioned? It needs context before introducing it.

We have removed the wildfires mention as it wasn't contributing to the overall message of the paper.

L41: The chemical composition, size and concentrations "of aerosols"

Fixed

L49: define PBL

Defined

L54: it does not need to be humid (which is not clearly defined in the first place). Just remove the "in a humid place".

Removed

L66: what is "p.201"?

Caused by a bug in the reference manager. IT has being addressed.

L66-67: give a ref for ELVOC and ULVOC. Additionally, their definition is not limited to 298K.

Fixed. It reads as: *"HOMs can be characterised as ultra-low VOCs (ULVOC) or extremely low VOCs (ELVOC) depending upon the saturation concentration (Bianchi et al., 2019; Peräkylä et al., 2020)."*

L68-71: rewrite.

Whole paragraph was rewritten as: *" The most common biogenic VOC (BVOC) is isoprene followed by monoterpenes. BVOCs play an important role in secondary organic aerosol (SOA) formation (e.g., Mahilang et al., 2021). VOCs have been associated with particle growth (Riipinen et al., 2012) but their role and the autoxidation mechanism was not understood until recently (Bianchi et al., 2019). Autoxidation of monoterpenes supports the particle growth process by generating highly oxygenated molecules (HOMs) via the formation of peroxy radicals (Bianchi et al., 2019; Kirkby et al., 2023; Lehtipalo et al., 2018). HOMs can be characterised as ultra-low VOCs (ULVOC) or extremely low VOCs (ELVOC) depending upon the saturation concentration (Bianchi et al., 2019; Peräkylä et al., 2020).*

*Oxidation of monoterpenes is a significant pathway for SOA formation, yielding higher amounts of low-volatility molecules like ULVOCs and ELVOCs compared to isoprene oxidation (Friedman and Farmer, 2018; Lee et al., 2023; Luo et al., 2024; Riva et al., 2019; Zhang et al., 2018). HOMs are key precursors for new particle formation. However, the atmospheric production of HOMs can be limited by competing reactions and the presence of other VOCs. For instance, as a general principle, once a VOC molecule oxidizes, it becomes more complex and forms larger Oxygenated VOCs (OVOCs) that are less likely to undergo further oxidation, especially in the presence of other VOCs with higher reactivity towards OH or $O_3$ (Kiendler-Scharr et al., 2009). An example of this limitation is the suppression of monoterpene-derived HOM formation by isoprene oxidation products. These products can interfere with the formation of $C_{20}$ dimers from monoterpene oxidation, leading to a reduced yield of HOMs and favoring the formation of weaker nucleating species $C_{15}$ (Dada et al., 2023; Heinritzi et al., 2020; Liu et al., 2016a). This suppression effect is dynamic, varying non-linearly with local atmospheric composition (e.g., isoprene and monoterpene concentrations, oxidant availability) and atmospheric conditions (e.g., temperature, humidity, stability), which ultimately determine the dominant SOA formation pathways (e.g. Song et al., 2019). "*

L71: Heiritzi et al. did not suggest isoprene oxidation products (C5) contribute to particle growth above 3 nm. Please check the paper again and rephrase.

Modified as shown in the previous comment.

L72-74: This sentence is a bit confusing. Nighttime NO3 oxidation is dominant and it needs not be compared with OH unless there is a strong OH source identified.

Modified as shown before.

L78: "Isoprene, monoterpenes, OH, nitrate radical and ozone availability"

Fixed the typo

L80: comma before "so"

Added a comma.

L83: C20 dimers "from monoterpene oxidation"

Fixed.

L115: a period before "On February" and a comma after.

Fixed.

L120: correct 5 February 5th.

Fixed.

L202: the subscription of vapours should be unified throughout the manuscript.

Fixed throughout the manuscript

L205: aerosol sulphate should be (SO4(2-)) and not SO4. Since this paper discusses gas phase radicals as well as particle phase compounds, it is advised to stick to coherent and annotations for these two categories throughout the manuscript, including captions such as Figure 2 etc.

We have changed SO4 to SO42- in both discussion and most captions.

L207-218: these discussions should be connected with an overview figure showing all the data. It can be presented in the SI for example. Since the cases are quite limited, it is a good idea to present a table giving a summary of all the events and what data each event has. So it would be easier to identify the events with the most complete data sets.

We have added Table 2 as a graphical way to briefly state what data is available on each event

L235-236: why Figure3 is mentioned before Figure2?

It wasn't. Figure 2 was introduced in line 201.

L247: why it increased? Could you give some discussions based on your data?

Added some discussions as follows: " …The first part of the regression shows slower growth rate. After the 6$^{th}$ hour of slow growth, the rate increases substantially, attributed to an increase of $H_2SO_4$ around this time. Shortly after this accelerated growth, there is a wind change from northerly to southerly (Figure S4). Following the southerly wind shift, a lower condensation sink and higher relative humidity likely contributed to the Gdp increase via enhanced condensation and water uptake. Declining tracer levels $SO_2$ and $NO_x$ indicate that local particle growth mechanisms were likely dominant over the influence of a new air mass up to the 7$^{th}$ hour when increases in $NO_x$ and $SO_2$ are observed. "

L252-255: rewrite

Rephrased as: "… These reaction pathways might include monoterpene ozonolysis and condensation over previously formed clusters (Liu et al., 2023; Wang et al., 2023), or oxygenated VOCs (OVOCs) brought to the site and condensed on formed seeds or may initiate nucleation (Bianchi et al., 2019; Higgins et al., 2022). Some of these processes were observed during the campaign and will be further explored on the nighttime events section. …"

L299: missing a period

Fixed.

L301: Kirkby et al should be referred to for the monoterpene nucleation[1].

Added Kirby et al (2023) as a reference here.

L302-303: could the authors give reasons why they believe the isoprene is contributing more to particle growth in their studies? I believe their instruments and methodology are not sufficient to derive such conclusion without detailed chemical modelling.

As suggested, we have rewritten much of the analysis in light of the references provided by the referee. The attribution to isoprene has been removed in most cases and it's only attributed to the growth phase (post NPF) under specific conditions.

L303-304: once again, how does the authors establish the connection of isoprene oxidation at the boundary layer with ELVOC without chemical information measured? The concentration of ELVOC by isoprene should be rather limited at this temperature. The authors should provide evidence to this. In fact, monoterpene concentration is also expected to increase by the same mechanism.

As stated in the previous comment, the discussion has been changed to reflect the literature.

L307: 1 ppb of monoterpene is substantial! The authors should check relevant papers such as Kirkby et al. and Lehtipalo et al.1,3 for quantitative assessment. This they should base their discussions on quantitative information provided in previous chamber experiments to further derive their conclusion. This is another piece of information that goes against their conclusion of significant isoprene contribution.

As stated in the previous comment, the discussion has been changed to reflect the literature.

L321: HOMS proxy or HOM proxy as written earlier? Again, HOMS or HOMs? Anyway, please be consistent with your writing and editing. Also note that throughout the manuscript, the subscriptions for $H_2SO_4$ etc has not been consistently applied. Please edit these details carefully.

Fixed throughout the text

L319-322: rewrite

Rephrased as: *"This is the first step in the reaction chain to produce $C_{15}$ dimers. This observation aligns with the HOM proxy (monoterpenes\*ozone): higher proxy values corresponded to periods of higher particle numbers, while a decrease in the HOM proxy coincided with a decrease in particle numbers and an increase in MACR + MVK products, suggesting a shift towards more isoprene-influenced atmospheric chemistry. Concurrently, increases in the organic and sulfate fractions, along with the condensation sink, indicate a shift towards conditions favoring the growth of existing larger particles through condensation and coagulation, rather than nucleation events."*

L343-344: again, please provide evidence of significant contribution of isoprene oxidation products in terms of previous laboratory experiments.

Rephrased to reflect the literature as: "The event on March 10 (see **Error! Reference source not found.**) shows high monoterpene concentrations that declined quickly just prior to the event being observed in the aerosol data. The aerosol growth phase is then observed to correlate with peaks in $SO_2$ and $NO_X$, as well as elevated levels of isoprene. Together, this suggests monoterpene ozonolysis initiated nucleation, followed by condensational growth via isoprene oxidation products."

L367-378: the discussions here are very poor. The choice of the referred paper is also rather problematic. The authors should check the conditions used in that experiments and whether that is really representative for the conditions in their site. For example, the experiments used $SO_2$ concentration of over 100 ppb and the conclusion about sCI and peroxides is rather odd. The main sink of $SO_2$ should either by OH or sCI.

We have removed all the discussion around $SO_2$ and humidity given that the conditions during the campaign are too different from the experiments and the results of those are only applicable under haze formation conditions.

L391-393: this is really a strong indication that the authors did not read Heinritzi paper at al.. The C15 is a product of monoterpene C10 and isoprene C5 and just from isoprene oxidation.. In fact, the Heinritzi paper clearly suggested that C15 is less able than monoterpene C20 in both nucleation and growth which the authors should take into account throughout the manuscript.

This was included in the SO2 humidity discussion that was removed from the manuscript.

L408: duplication period.

fixed

L413 on: In my previous comment on that the statement "MACR is oxidised to heavier OVOCs that eventually condense." is wrong, the authors used Heinritzi et al. 2020 as the support of their statement which was totally non relevant to the discussion. The problem is that MACR is a four carbon species which is not believed to be the major precursor of condensable isoprene molecules (C5 species). I refer the authors to Wennberg et al.4 for some basic knowledge of isoprene oxidation mechanisms.

This discussion was changed to reflect current literature and Australian conditions as follows: " Australia experiences an isoprene-dominated atmosphere (Emmerson et al., 2016; Ramirez-Gamboa et al., 2021), and the chemical balance in the atmosphere can rapidly change, particularly in the hotter seasons when more isoprene is emitted. While SOA formation on pre-existing particles can involve molecules with relatively high saturation vapor pressures, new particle formation critically depends on molecules with extremely low saturation vapor pressures due to the Kelvin effect (Tröstl et al., 2016). Heinritzi et al. (2020) showed that reducing $C_{20}$ formation (α-pinene oxidation in isoprene presence) to favor $C_{15}$ formation reduces nucleation rates. However, it is also important to highlight that $C_{15}$, $C_{10}$, and even $C_5$ oxidation products from isoprene oxidation can contribute to SOA mass on existing particles. Therefore, in Australia's isoprene-dominated environment, higher isoprene to monoterpene ratios could lead to a greater production of $C_5$ and $C_{15}$ products that contribute to particle growth on existing aerosols (and SOA mass), while simultaneously hindering new particle formation by reducing the formation of $C_{20}$ dimers from monoterpenes. "

Figure 9 is missing a legend for particles.

Fixed

L440-441: isoprene oxidation by nitrate pathway also produces MVK and MACR4. Could this discussion be more quantitative. Additionally, from the figure the decrease of isoprene seems not relevant to the NOx increase. So why the authors believe that isoprene has a role here?

This discussion has been significantly revised and now reads as follows:

"When VOC data are available, monoterpene concentrations were moderate and increased steadily during the event (5th Feb and 9th March). Isoprene was high at the start of the event on 5th Feb, (see Figure 8) however the sudden decrease in isoprene concentration likely coincides with sunset on that day.  When aerosol composition data was available (10th Feb) aerosol organic, nitrate and sulphate concentrations increase during the event. When ozone data were available, concentrations decreased slightly during the course of the event.

The frequency of nocturnal events observed in this study is lower than observed previously at a nearby location (Tumbarumba a eucalypt forest site located 300 km to the SE of Cataract (Suni et al., 2009)), where in the summer of 2006, nocturnal NPF events were observed on 32% of the analysed nights and occurred 2.5 times more frequently than daytime events.  Simulating the NPF at Tumbarumba, Ortega et al (2012) was able to reproduce the observations from Tumbarumba by ozonolysis of 13 -

carene to initiate nucleation and a-pinene to grow particle diameters. Ozonolysis of limonene was found to contribute to both nucleation and aerosol growth.  The lower frequency observed in our study may be linked to the apparent initiations of nucleation by $NO_2$, which nocturnally can react with $O_3$ to form nitrate radicals.   Li et al. (2024), suggest even trace amounts of $NO_3$ radicals suppress the NPF."

---

## Author Response (AR3)

We sincerely appreciate the diligent effort from the reviewers and editors in their assessment of this manuscript.

Regarding the comment pertaining to **line 429**, we acknowledge that this was a typographical error. The text has now been corrected to accurately read "**inhibition**" instead of "initiation."